# COVID19 Disease Map, a computational knowledge repository of virus–host interaction mechanisms

Marek Ostaszewski[1,*], Anna Niarakis[2,3], Alexander Mazein[1], Inna Kuperstein[4,5,6], Robert Phair[7], Aurelio Orta-Resendiz[8,9], Vidisha Singh[2], Sara Sadat Aghamiri[10], Marcio Luis Acencio[1], Enrico Glaab[1], Andreas Ruepp[11], Gisela Fobo[11], Corinna Montrone[11], Barbara Brauner[11], Goar Frishman[11], Luis Cristóbal Monraz Gómez[4,5,6], Julia Somers[12], Matti Hoch[13], Shailendra Kumar Gupta[13], Julia Scheel[13], Hanna Borlinghaus[14], Tobias Czauderna[15], Falk Schreiber[14,15], Arnau Montagud[16], Miguel Ponce de Leon[16], Akira Funahashi[17], Yusuke Hiki[17], Noriko Hiroi[18], Takahiro G Yamada[17], Andreas Dräger[19,20,21], Alina Renz[19,20], Muhammad Naveez[13,22], Zsolt Bocskei[23], Francesco Messina[24,25], Daniela Börnigen[26], Liam Fergusson[27], Marta Conti[28], Marius Rameil[28], Vanessa Nakonecnij[28], Jakob Vanhoefer[28], Leonard Schmiester[28,29], Muying Wang[30], Emily E Ackerman[30], Jason E Shoemaker[30,31], Jeremy Zucker[32], Kristie Oxford[32], Jeremy Teuton[32], Ebru Kocakaya[33], Gökçe Yağmur Summak[33], Kristina Hanspers[34], Martina Kutmon[35,36], Susan Coort[35], Lars Eijssen[35,37], Friederike Ehrhart[35,37], Devasahayam Arokia Balaya Rex[38], Denise Slenter[35], Marvin Martens[35], Nhung Pham[35], Robin Haw[39], Bijay Jassal[39], Lisa Matthews[40], Marija Orlic-Milacic[39], Andrea Senff Ribeiro[39,41], Karen Rothfels[39], Veronica Shamovsky[40], Ralf Stephan[39], Cristoffer Sevilla[42], Thawfeek Varusai[42], Jean-Marie Ravel[43,44], Rupsha Fraser[45], Vera Ortseifen[46], Silvia Marchesi[47], Piotr Gawron[1,48], Ewa Smula[1], Laurent Heirendt[1], Venkata Satagopam[1], Guanming Wu[49], Anders Riutta[34], Martin Golebiewski[50], Stuart Owen[51], Carole Goble[51], Xiaoming Hu[50], Rupert W Overall[52,53,54], Dieter Maier[55], Angela Bauch[55], Benjamin M Gyori[56], John A Bachman[56], Carlos Vega[1], Valentin Grouès[1], Miguel Vazquez[16], Pablo Porras[42], Luana Licata[57], Marta Iannuccelli[57], Francesca Sacco[57], Anastasia Nesterova[58], Anton Yuryev[58], Anita de Waard[59], Denes Turei[60], Augustin Luna[61,62], Ozgun Babur[63], Sylvain Soliman[3], Alberto Valdeolivas[60], Marina Esteban-Medina[64,65], Maria Peña-Chilet[64,65,66], Kinza Rian[64,65], Tomáš Helikar[67], Bhanwar Lal Puniya[67], Dezso Modos[68,69], Agatha Treveil[68,69], Marton Olbei[68,69], Bertrand De Meulder[70], Stephane Ballereau[71], Aurélien Dugourd[60,72], Aurélien Naldi[3], Vincent Noël[4,5,6], Laurence Calzone[4,5,6], Chris Sander[61,62], Emek Demir[12], Tamas Korcsmaros[68,69], Tom C Freeman[73], Franck Augé[23], Jacques S Beckmann[74], Jan Hasenauer[75,76], Olaf Wolkenhauer[13], Egon L Wilighagen[35], Alexander R Pico[34], Chris T Evelo[35,36], Marc E Gillespie[39,77], Lincoln D Stein[39,78], Henning Hermjakob[42], Peter D'Eustachio[40], Julio Saez-Rodriguez[60], Joaquin Dopazo[64,65,66,79], Alfonso Valencia[16,80], Hiroaki Kitano[81,82], Emmanuel Barillot[4,5,6], Charles Auffray[71], Rudi Balling[1], Reinhard Schneider[1] & the COVID-19 Disease Map Community[†]

---

1–82 The list of affiliations appears at the end of this article
*Corresponding author. Tel: +352 46 66 44 5604; E-mail: marek.ostaszewski@uni.lu
†FAIRDOMHub: https://fairdomhub.org/projects/190

---

# Abstract

We need to effectively combine the knowledge from surging literature with complex datasets to propose mechanistic models of SARS-CoV-2 infection, improving data interpretation and predicting key targets of intervention. Here, we describe a large-scale community effort to build an open access, interoperable and computable repository of COVID-19 molecular mechanisms. The COVID-19 Disease Map (C19DMap) is a graphical, interactive representation of disease-relevant molecular mechanisms linking many knowledge sources. Notably, it is a computational resource for graph-based analyses and disease modelling. To this end, we established a framework of tools, platforms and guidelines necessary for a multifaceted community of biocurators, domain experts, bioinformaticians and computational biologists. The diagrams of the C19DMap, curated from the literature, are integrated with relevant interaction and text mining databases. We demonstrate the application of network analysis and modelling approaches by concrete examples to highlight new testable hypotheses. This framework helps to find signatures of SARS-CoV-2 predisposition, treatment response or prioritisation of drug candidates. Such an approach may help deal with new waves of COVID-19 or similar pandemics in the long-term perspective.

**Keywords** computable knowledge repository; large-scale biocuration; omics data analysis; open access community effort; systems biomedicine
**Subject Categories** Computational Biology; Microbiology, Virology & Host Pathogen Interaction
**Mol Syst Biol. (2021) 17: e10387**

# Introduction

The coronavirus disease 2019 (COVID-19) pandemic due to severe acute respiratory syndrome coronavirus 2 (SARS-CoV-2) has already resulted in the infection of over 250 million people worldwide, of whom almost 5 million have died (https://covid19.who.int, accessed on 05.10.2021). This global challenge motivated researchers worldwide to an unprecedented effort towards understanding the pathology to treat and prevent it. To date, over 170 thousand articles have been published in relation to COVID-19 (PubMed query "covid-19[Title/Abstract] or sars-cov-2[Title/Abstract]", accessed on 01.07.2021). The reported molecular pathophysiology that links SARS-CoV-2 infection to the clinical manifestations and course of COVID-19 is complex and spans multiple biological pathways, cell types and organs (Gagliardi *et al*, 2020). Resources such as Protein Data Bank repository of viral protein structures (preprint: Lubin *et al*, 2020) or the IMEx coronavirus interactome (Perfetto *et al*, 2020) offer detailed information about particular viral proteins and their direct binding partners. However, the scope of this information is limited. To gain insight into the large network of molecular mechanisms, knowledge from the vast body of scientific literature and bioinformatic databases needs to be integrated using systems biology standards. A repository of such computable knowledge will support data analysis and predictive modelling.

With this goal in mind, we initiated a collaborative effort involving over 230 biocurators, domain experts, modellers and data analysts from 120 institutions in 30 countries to develop the COVID-19 Disease Map (C19DMap), an open access collection of curated computational diagrams and models of molecular mechanisms implicated in the disease (Ostaszewski *et al*, 2020). The C19DMap is a constantly evolving resource, refined and updated by ongoing biocuration, sharing and analysis efforts. Currently, it is a collection of 42 diagrams containing 1,836 interactions between 5,499 elements, supported by 617 publications and preprints. The summary of diagrams available in the C19DMap can be found online (https://covid.pages.uni.lu/map_contents) and in Table EV1.

In the article, we explain the effort of our multidisciplinary community to construct the interoperable content of the resource, involving biocurators, domain experts and data analysts. We introduce the scope of the C19DMap and the insight it brings into the crosstalk and regulation of COVID-19-related molecular mechanisms. Next, we outline analytical workflows that can be used on the contents of the map, including the initial outcomes of two case studies. We conclude with a discussion on the utility and perspectives of the C19DMap as a disease-relevant computational repository.

# Results

### An interoperable repository of comprehensive and computable diagrams

We constructed a comprehensive diagrammatic description of disease mechanisms in a way that is both human- and machine-readable, lowering communication barriers between experimental and computational biologists. To this end, we aligned the biocuration efforts of the Disease Maps Community (Mazein *et al*, 2018), Reactome (Jassal *et al*, 2020), and WikiPathways (Slenter *et al*, 2018) and developed guidelines for building and annotating these diagrams. In addition, we integrated relevant knowledge from public repositories (Licata *et al*, 2020; Perfetto *et al*, 2020; Rodchenkov *et al*, 2020; Türei *et al*, 2021) and text mining resources to update and refine the contents of the C19DMap based on other knowledge-building efforts. This work resulted in a series of pathway diagrams constructed *de novo*, describing key events in the COVID-19 infectious cycle and host response.

The C19DMap project involved three main groups of participants: the biocurators, the domain experts, and the analysts and modellers. Biocurators developed a collection of systems biology diagrams focused on the molecular mechanisms of SARS-CoV-2. Domain experts refined the contents of the diagrams using interactive visualisation and annotations. Analysts and modellers developed computational workflows to generate hypotheses and predictions about the mechanisms encoded in the diagrams. Figure 1 illustrates the ecosystem of the C19DMap Community, highlighting the roles of the participants, available format conversions, interoperable tools and downstream uses. The community members and their contributions are listed on FAIRDOMHub (Wolstencroft *et al*, 2017).

### *Creating and accessing the diagrams*

The biocurators of the C19DMap diagrams followed the guidelines developed by the Community, WikiPathways (Slenter *et al*, 2018) and Reactome (Jassal *et al*, 2020) based on systems biology

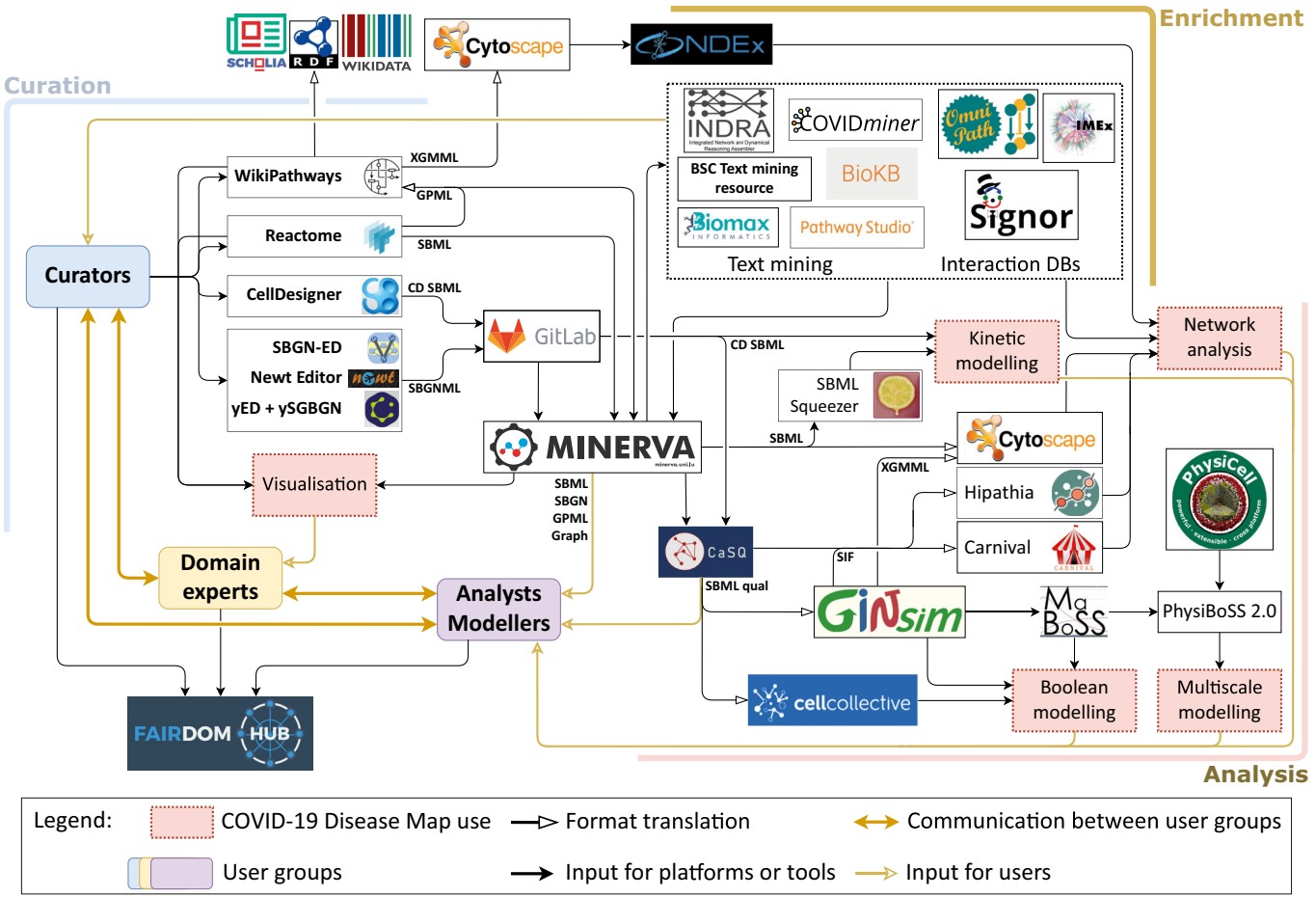

**Figure 1. Ecosystem of the COVID-19 Disease Map Community.**

The main groups of the C19DMap Community are biocurators, domain experts, and analysts and modellers; communicating to refine, interpret, and apply C19DMap diagrams. These diagrams are created and maintained by biocurators, following pathway database workflows or stand-alone diagram editors, and reviewed by domain experts. The content is shared via pathway databases or a GitLab repository; all can be enriched by integrated resources of text mining and interaction databases. The C19DMap diagrams are available in several layout-aware systems biology formats and integrated with external repositories, allowing a range of computational analyses, including network analysis and Boolean, kinetic or multiscale simulations.

standards (Le Novère *et al*, 2009; Demir *et al*, 2010; Keating *et al*, 2020) and persistent identifiers (Wimalaratne *et al*, 2018). The diagrams are composed of biochemical reactions and interactions (altogether called interactions) between different molecular entities in various cellular compartments. As multiple teams worked on related topics, biocurators reviewed other diagrams, also across platforms (see also Materials and Methods). The diagrams are accessible online and can be explored using an intuitive user interface. Table 1 summarises information about the curated diagrams, and Table EV1 lists the diagrams and provides links to access them.

*Enrichment using knowledge from databases and text mining*
The knowledge of COVID-19 mechanisms is rapidly evolving, as shown by the growth of the COVID-19 Open Research Dataset (CORD-19), a source of manuscripts and metadata on COVID-19-related research (preprint: Lu Wang *et al*, 2020). CORD-19 currently contains almost 480,000 articles and preprints, over ten times more than when it was introduced more than a year ago (accessed on 05.10.2021). In such a quickly evolving environment, manual

biocuration needs to be supported by automated procedures to identify and prioritise crucial articles, molecules and their interactions to be included in the C19DMap.

Potential knowledge sources for such assisted biocuration are interaction and pathway databases, especially those with dedicated COVID-19 content (Licata *et al*, 2020; Perfetto *et al*, 2020). Their structured and annotated information on protein interactions or causal relationships was generated using separate biocuration guidelines and formats. Nevertheless, their comparable identifiers and references to source publications make them plausible building blocks for constructing the C19DMap (see Materials and Methods).

Text mining approaches are another source of information that can direct the biocurators towards the most recent and relevant findings. They automatically extract and annotate biomolecule names and their interactions from abstracts, full-text documents or pathway figures (Bauch *et al*, 2020; Hanspers *et al*, 2020). Networks of molecule interactions constructed by text mining can carry substantially more noise than the contents of interaction databases but offer broader literature coverage.

**Table 1. COVID-19 Disease Map contents.**

| | Source | | |
|---|---|---|---|
| | Individual diagrams | Reactome | WikiPathways |
| Diagram contents | 21 diagrams<br>1,334 interactions<br>4,272 molecular entities<br>397 publications | 2 diagrams<br>101 interactions<br>489 molecular entities<br>227 publications | 19 diagrams<br>401 interactions<br>738 molecular entities<br>61 publications |
| Access | GitLab<br>gitlab.lcsb.uni.lu/covid/models | SARS-CoV-1 and SARS-CoV-2 infections collection<br>reactome.org/PathwayBrowser/#/<br>R-HSA-9679506 | COVID pathway collection<br>covid.wikipathways.org |
| Exploration | The MINERVA Platform<br>(Gawron et al, 2016)<br>covid19map.elixir-luxembourg.org<br>Guide: covid.pages.uni.lu/minerva-guide | Native web interface<br>Guide: covid.pages.uni.lu/reactome-guide | Native web interface<br>Guide: covid.pages.uni.lu/wikipathways-guide |
| Biocuration guidelines | Community[a] | Platform-specific[b] | Platform-specific[c] |
| Diagram Editors | CellDesigner (Matsuoka et al, 2014),<br>Newt[d], SBGN-ED (Czauderna et al,<br>2010), yEd+ySBGN[e] | Reactome pathway editor[b] | PathVisio<br>(Kutmon et al, 2015) |
| Formats | CellDesigner SBML<br>SBGNML (Bergmann et al, 2020) | Internal, SBML and SBGNML compliant | GPML (Kutmon et al, 2015) |

The table summarises biocuration resources and content of the C19DMap across three main parts of the repository. All diagrams are listed in Table EV1, and available online at: https://covid.pages.uni.lu/map_contents.
[a]https://fairdomhub.org/documents/661
[b]https://reactome.org/community/training
[c]https://www.wikipathways.org/index.php/Help:Editing_Pathways
[d]https://newteditor.org
[e]https://github.com/sbgn/ySBGN

Table 2 summarises open access interaction databases and text mining knowledge bases supporting the biocuration of the C19DMap. Molecular interactions from these sources have a broad coverage at the cost of depth of mechanistic representation. The biocurators used this content to build and update the map by manual exploration or by programmatic comparison. First, the biocurators visually explored the contents of such networks using available search interfaces to identify interactions of interesting molecules and encoded them in the diagrams. This task was supported by a dedicated visualisation tool COVIDminer (https://rupertoverall.net/covidminer). The biocurators also used assistant chatbots that respond to natural language queries and return

**Table 2. Resources supporting biocuration of the COVID-19 Disease Map.**

| Resource | Type | Manually curated | Directed | Layout | COVID-19 specific |
|---|---|---|---|---|---|
| IMEx Consortium (Orchard et al, 2012) | Interaction database | Yes | No | No | Yes[a] |
| SIGNOR 2.0 (Licata et al, 2020) | | Yes | Yes | Yes | Yes[b] |
| OmniPath (Türei et al, 2016) | | No | Yes | No | No |
| Elsevier Pathway Collection[c] | Pathway | Yes | Yes | Yes | Yes[d] |
| INDRA (Gyori et al, 2017) | Text mining | Yes | Yes | No | Yes[e] |
| BioKB[f] | | No | Yes | No | Yes |
| AILANI COVID-19[g] | | No | Yes | No | Yes |
| OpenNLP+GNormPlus[h] | | No | Yes | No | Yes |

They include (i) collections of COVID-19 interactions and pathways, (ii) interaction databases and (iii) text mining corpora.
[a]https://www.ebi.ac.uk/intact/resources/datasets#coronavirus
[b]https://signor.uniroma2.it/covid/
[c]https://pathwaystudio.com
[d]http://dx.doi.org/10.17632/d55xn2c8mw.1
[e]https://emmaa.indra.bio/dashboard/covid19
[f]https://biokb.lcsb.uni.lu
[g]https://ailani.ai
[h]https://gitlab.lcsb.uni.lu/covid/models/-/tree/master/Resources/Text%20mining

meaningful answers extracted from text mining platforms. Second, we developed routines for programmatic queries of the interaction databases, providing automated and reproducible exploration of the selected databases. This was realised using data endpoints: Application programming interfaces (API) for INDRA, AILANI or Pathway Studio, and SPARQL for BioKB. This automated exploration retrieved functions, interactions, pathways or drugs associated with submitted queries, e.g. gene lists. This way, otherwise time-consuming tasks such as an assessment of completeness of a given diagram or search for new literature evidence were automated. Section "Exploration of the networked knowledge" describes an application of such automated queries in crosstalk analysis.

### Interoperability of the diagrams and annotations

The biocuration of the C19DMap diagrams was distributed across multiple teams, using varying tools and associated systems biology representations. This required a common approach to annotations of diagram elements and their interactions. Additionally, to compare and combine the diagrams in the C19DMap, interoperability of layout-aware formats was needed.

The diagrams were encoded in three layout-aware formats for standardised representation of molecular interactions: SBML, SBGNML and GPML. All three formats, centred around molecular interactions, provided a constrained vocabulary to encode element and interaction types, encode layout of corresponding diagrams and support stable identifiers for diagram components. These shared

properties, supported by a common ontology (Courtot *et al*, 2011), allowed cross-format translation of the diagrams, which was essential for harmonising the effort between biocuration platforms.

The ecosystem of tools and resources supporting the C19DMap (see Fig 1) ensured interoperability between SBML, SBGNML and GPML via translation, preserving the diagram layout (Bohler *et al*, 2016; Balaur *et al*, 2020; Hoksza *et al*, 2020) for harmonised visualisation of diagrams. Additionally, these diagrams were transformed into inputs of computational pipelines and data repositories, allowing network analysis, pathway modelling and interoperability with molecular interaction repositories (Pillich *et al*, 2017) (see Materials and Methods).

### Structure and scope of the COVID-19 Disease Map

The C19DMap was built bottom-up, exploiting a rich bioinformatics framework discussed in Section "An interoperable repository of comprehensive and computable diagrams" of the Results, based on knowledge from existing studies of other coronaviruses (Fung & Liu, 2019) and contextualised with data emerging from studies of SARS-CoV-2 (Gordon *et al*, 2020). The contents of the C19DMap are available online, summarised in a constantly updated overview at https://covid.pages.uni.lu/map_contents (see also Table EV1). Currently, the C19DMap focuses on molecular processes involved in SARS-CoV-2 entry and replication and host–virus interactions (see Fig 2). Emerging scientific evidence of host susceptibility, immune

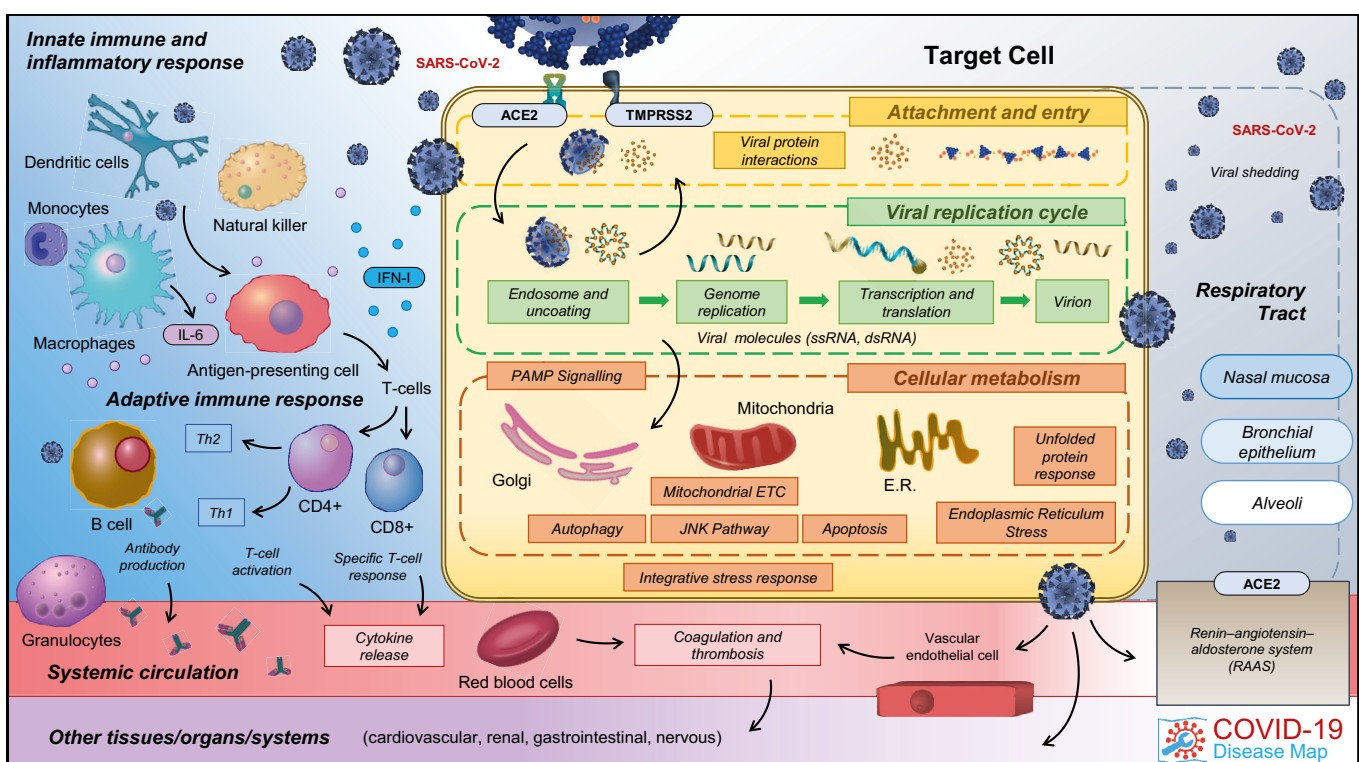

**Figure 2.  The structure and content of the COVID-19 Disease Map.**

An overview of the areas of focus of the C19DMap biocuration. Go to covid19map.elixir-luxembourg.org for an interactive version. Full list of diagrams and browsing instructions are available online at covid.pages.uni.lu/map_contents.

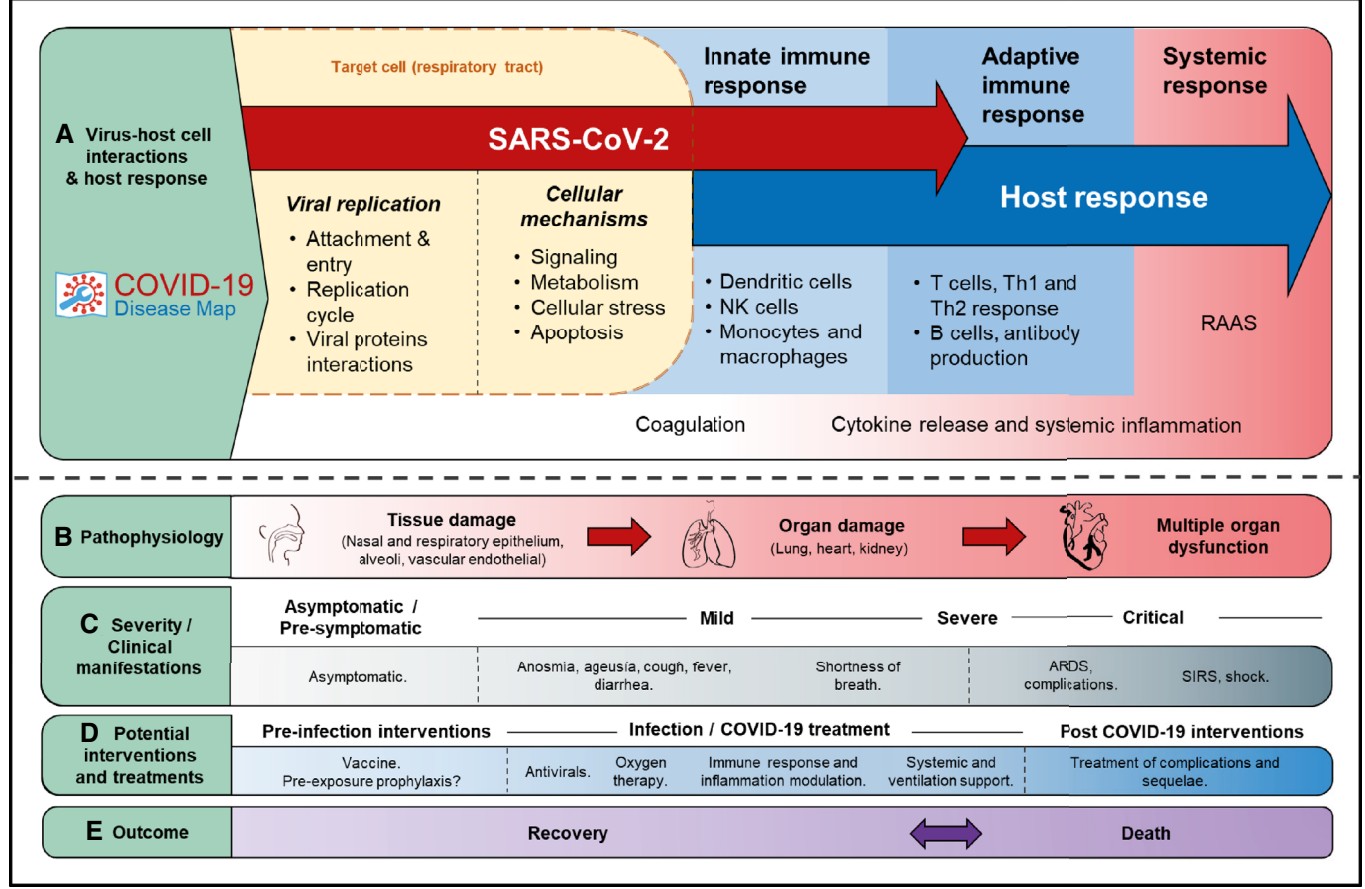

**Figure 3. Overview of the C19DMap in the context of COVID-19 progression.**

The figure summarises the main sections and content of the C19DMap by illustrating the progressive but overlapping mechanisms at different levels and study features of the disease intended as quick references for the map.

A  Cellular level (light yellow), the immune response (blue) and other systemic responses (red) of the host following SARS-CoV-2 infection.

B  The progression of pathophysiology from tissue damage to organ damage and multiple organ dysfunction in severe cases.

C  Clinical manifestations, depending on the severity of the infection from asymptomatic to critical COVID-19.

D  Potential intervention strategies that may be suggested based on the analysis of the C19DMap before, during and after infection, depending on the type and target of the intervention.

E  Clinical outcomes of SARS-CoV-2 infection. ARDS, acute respiratory distress syndrome. RAAS, renin–angiotensin–aldosterone system. SIRS, systemic inflammatory response syndrome. For the literature on clinical manifestations, see Lauer *et al*, 2020; He *et al*, 2020; Huang *et al*, 2020; Bajema *et al*, 2020; preprint: Chen *et al*, 2020b; Wang *et al*, 2020a; Tong *et al*, 2020.

response, cell and organ specificity will be incorporated into the next versions in accordance with our curation roadmap (https://fairdomhub.org/documents/907).

While the interactions of SARS-CoV-2 with various host cell types are vital determinants of COVID-19 pathology (Hui *et al*, 2020; Mason, 2020; Ziegler *et al*, 2020), the current C19DMap represents an infection of a generic host cell. Several pathways included in the map are shared between different cell types; for example, the IFN-1 pathway is active in dendritic and lung epithelial cells and in alveolar macrophages (Hadjadj *et al*, 2020; Lee & Shin, 2020; Sa Ribero *et al*, 2020). Continued annotations of emerging expression datasets (Delorey *et al*, 2021) and other sources of information will allow the construction of cell-specific versions of the C19DMap to provide an integrated view of the effects of SARS-CoV-2 on the human organism. An example workflow to construct such a focused version of

the map was proposed in Section "Case study: analysis of cell-specific mechanisms using single-cell expression data".

SARS-CoV-2 infection and COVID-19 progression are sequential events that start with viral attachment and entry (Fig 3). These events involve various dynamic processes and different timescales that are not captured in static representations of pathways. The correlation of symptoms and potential drugs suggested to date helps downstream data exploration and drug target interpretation in the context of therapeutic interventions.

### Contents of the COVID-19 Disease Map
#### Virus replication cycle

The virus replication cycle includes binding of the spike surface glycoprotein (S) to angiotensin-converting enzyme 2 (ACE2) facilitated by TMPRSS2 (Hoffmann *et al*, 2020b; Letko *et al*, 2020) and

other receptors (preprint: Amraei *et al*, 2020; preprint: Gao *et al*, 2020). Viral entry occurs either by direct fusion of the virion with the cell membranes or by endocytosis (Hoffmann *et al*, 2020a; Xia *et al*, 2020) of the virion membrane and the subsequent injection of the nucleocapsid into the cytoplasm. Within the host cell, the C19DMap depicts how SARS-CoV-2 hijacks the rough endoplasmic reticulum (RER)-linked host translational machinery for its replication (Chen *et al*, 2010; Angelini *et al*, 2013; Nakagawa *et al*, 2016; V'kovski *et al*, 2019). The RER-attached translation machinery produces structural proteins, which, together with the newly generated viral RNA, are assembled into new virions and released to the extracellular space via smooth-walled vesicles (Nakagawa *et al*, 2016) or hijacked lysosomes (Ghosh *et al*, 2020).

These mechanisms are illustrated in the diagrams of the "Virus replication cycle" section in Table EV1: "Attachment and entry", "Transcription, translation and replication" and "Assembly and release".

### Viral subversion of host defence

Endoplasmic reticulum (ER) stress results from the production of large amounts of viral proteins that create an overload of unfolded proteins (Krähling *et al*, 2009; DeDiego *et al*, 2011; Fukushi *et al*, 2012). The mechanisms of the unfolded protein response (UPR) include the mitigation of the misfolded protein load by reduced protein synthesis and increased protein degradation (Sureda *et al*, 2020) through the ubiquitin–proteasome system (UPS) and autophagy (Choi *et al*, 2018; Bello-Perez *et al*, 2020). SARS-CoV-2 may perturb the process of UPS-based protein degradation via the interaction of the viral Orf10 protein with the Cul2 ubiquitin ligase complex and its putative substrates (Gordon *et al*, 2020; Zhang *et al*, 2020). The involvement of SARS-CoV-2 in autophagy is less documented (Yang & Shen, 2020).

The increased burden of misfolded proteins due to viral replication and subversion of mitigation mechanisms may trigger programmed cell death (apoptosis). The C19DMap encodes major signalling pathways triggering this final form of cellular defence against viral replication (Diemer *et al*, 2010). Many viruses block or delay cell death by expressing anti-apoptotic proteins to maximise the production of viral progeny (Kanzawa *et al*, 2006; Liu *et al*, 2007) or induce it in selected cell types (Diemer *et al*, 2010; Chu *et al*, 2016; preprint: Chen *et al*, 2020b).

These mechanisms are illustrated in the diagrams of the "Viral subversion of host defence" section in Table EV1: "ER stress and unfolded protein response", "Autophagy and protein degradation" and "Apoptosis".

### Host integrative stress response

Severe acute respiratory syndrome coronavirus 2 infection damages the epithelium and the pulmonary capillary vascular endothelium (Bao *et al*, 2020), impairing respiration and leading to acute respiratory distress syndrome (ARDS) in severe forms of COVID-19 (Huang *et al*, 2020). The release of pro-inflammatory cytokines and hyperinflammation are known complications, causing further widespread damage (Chen *et al*, 2020a; Lucas *et al*, 2020). Coagulation disturbances and thrombosis are associated with severe cases, but specific mechanisms have not been described yet (Iba *et al*, 2020; Klok *et al*, 2020). Nevertheless, it was shown that SARS-CoV-2 disrupts the coagulation cascade and causes renin–angiotensin system (RAS) imbalance (Magro *et al*, 2020; Urwyler *et al*, 2020).

Angiotensin-converting enzyme 2, used by SARS-CoV-2 for host cell entry, is a regulator of RAS and is widely expressed in the affected organs. The diagrams in the repository describe how ACE2-converted angiotensins trigger the counter-regulatory arms of RAS and the downstream signalling via AGTR1, regulating the coagulation cascade (Gheblawi *et al*, 2020; McFadyen *et al*, 2020).

These mechanisms are illustrated in the diagrams of the "Integrative stress response" section in Table EV1: "Renin–angiotensin system" and "Coagulopathy".

### Host immune response

The innate immune system detects specific pathogen-associated molecular patterns, through pattern recognition receptors (PRRs) that recognise viral RNA in the endosome during endocytosis or in the cytoplasm during virus replication. The PRRs activate associated transcription factors promoting the production of antiviral proteins such as interferon-alpha, interferon-beta and interferon-lambda (Takeuchi & Akira, 2010; Berthelot & Lioté, 2020; Blanco-Melo *et al*, 2020; Hadjadj *et al*, 2020; Park & Iwasaki, 2020). SARS-CoV-2 impairs this mechanism (Chu *et al*, 2020), but the exact components are yet to be elucidated (Liao *et al*, 2005; Devaraj *et al*, 2007; Frieman *et al*, 2007; Li *et al*, 2016; Bastard *et al*, 2020). The C19DMap includes both the virus recognition process and the viral evasion mechanisms. It provides the connection between virus entry, its replication cycle, and the effector pathways of pro-inflammatory cytokines, especially of the interferon type I cascade (Wong *et al*, 2018; Mesev *et al*, 2019; Mantlo *et al*, 2020; Su & Jiang, 2020; Thoms *et al*, 2020; Ziegler *et al*, 2020).

Key metabolic pathways modulate the availability of nutrients and critical metabolites of the immune microenvironment (Rao *et al*, 2019). They are a target of infectious agents that reprogram host metabolism to create favourable conditions for their reproduction (Kedia-Mehta & Finlay, 2019). The C19DMap encodes several immunometabolic pathways and provides detailed information about the way SARS-CoV-2 proteins interact with them. The metabolic pathways include haem catabolism (Batra *et al*, 2020) and its downstream target, the NLRP3 inflammasome (van den Berg & Te Velde, 2020), tryptophan-kynurenine metabolism governing the response to inflammatory cytokines (Murakami *et al*, 2013; preprint: Su *et al*, 2020), and nicotinamide and purine metabolism (Renz *et al*, 2020). Finally, we represent the pyrimidine synthesis pathway, tightly linked to purine metabolism, affecting viral DNA and RNA syntheses (Hayek *et al*, 2020; Xiong *et al*, 2020).

These mechanisms are illustrated in the diagrams of the "Innate Immune Response" section in Table EV1: "PAMP signalling", "Induction of interferons and the cytokine storm" and "Altered host metabolism".

### Exploration of the networked knowledge

The diagrams of the C19DMap were curated in a distributed manner across various platforms and tools. In order to coordinate such an effort and get a systematic overview of the contents of the map, we programmatically analysed the content of the diagrams, benefiting from their standard encoding and annotation (see Materials and Methods). This allowed us to identify crosstalk and functional overlaps across pathways. Then, we linked the diagrams to interaction and text mining databases to fill the gaps in our understanding of COVID-19 mechanisms and generate new testable hypotheses.

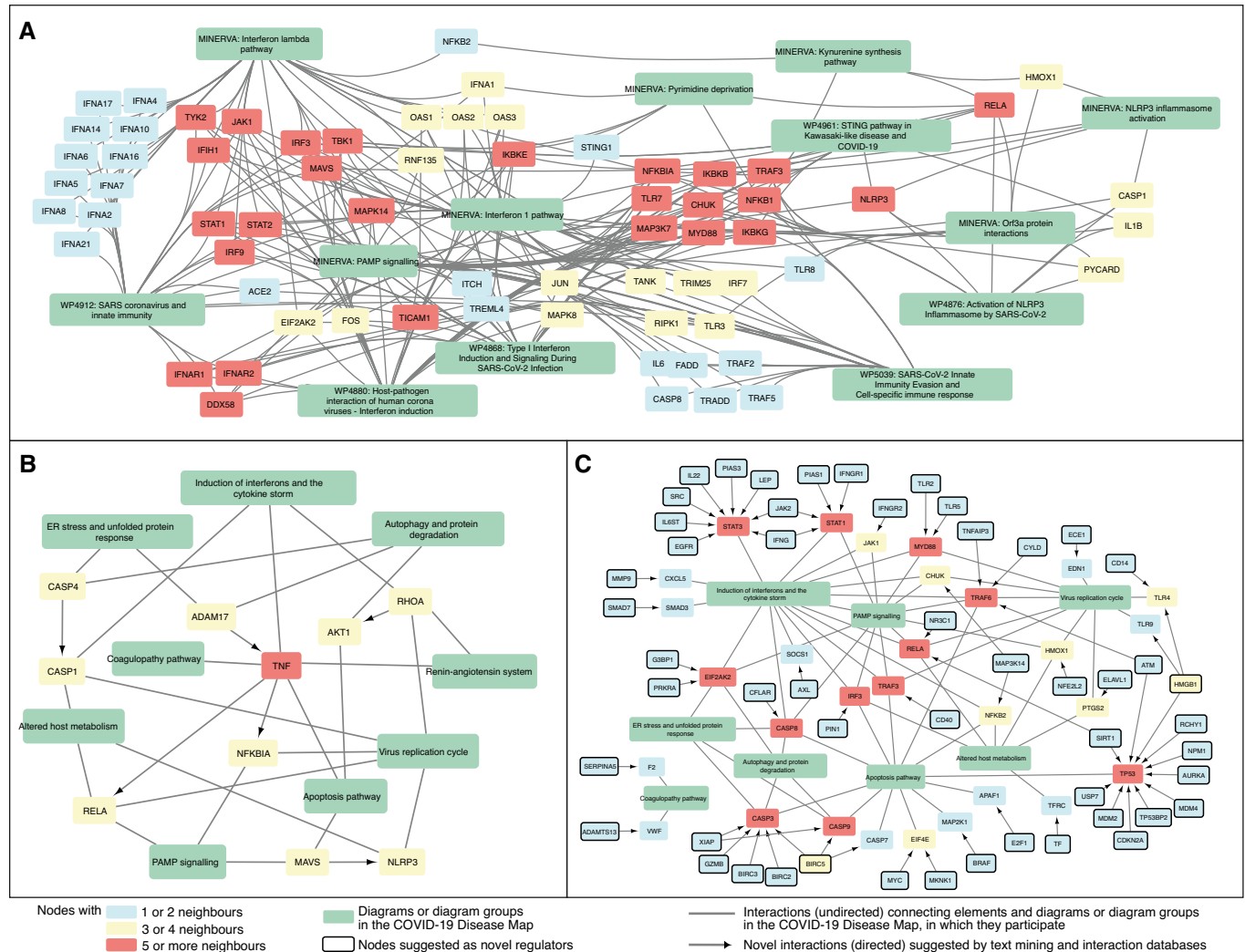

**Figure 4. Exploration of the existing and candidate crosstalk between the diagrams of the COVID-19 Disease Map.**

The network structure of the diagrams and their interactions based on existing crosstalk (shared elements), candidate crosstalk, and candidate regulators. Colour code: green—pathways or pathway groups; blue—proteins with one or two neighbours; yellow—proteins with three or four neighbours; and red—proteins with five or more neighbours. Candidate molecular interactions are shown as directed edges. Candidate regulator elements are marked with a solid black border. See Materials and Methods for details.

A  Existing crosstalk between individual diagrams of IFN-I and RELA-related mechanisms.
B  Candidate crosstalk between pathway groups.
C  Candidate regulators of existing diagrams from text mining and interaction databases.

Below, we discuss three specific examples of exploration of this networked knowledge (see Fig 4). Access to the complete content of the crosstalk diagrams can be found in Materials and Methods.

### Existing crosstalk between COVID-19 Disease Map diagrams

First, the existing pathways crosstalk emerged by matching entities between the diagrams (Figs 4A, EV1 and EV2). For instance, they link different pathways involved in type I IFN (IFN-1) signalling. Responses to RNA viruses and pathogen-associated molecular patterns (PAMPs) share common pathways, involving RIG-I/Mda-5, TBK1/IKKE and TLR signalling, leading to the production of IFN-1s, especially IFN-beta (Häcker & Karin, 2006) and IFN-alpha

(Mogensen, 2009). Downstream, IFN-1 activates Tyk2 and Jak1 protein tyrosine kinases, causing STAT1:STAT2:IRF9 (ISGF3) complex formation to promote the transcription of IFN-stimulated genes (ISGs). Importantly, TBK1 also phosphorylates IKBA, an NF-kB inhibitor, for proteasomal degradation in crosstalk with the UPS pathway, allowing free NF-kB and IRF3 to co-activate ISGs (Fang et al, 2017). Another TBK1 activator, STING, links IFN signalling with pyrimidine metabolism.

SARS-CoV-2 M protein affects these IFN responses by inhibiting the RIG-I:MAVS:TRAF3 complex and TBK1, preventing IRF3 phosphorylation, nuclear translocation and activation (Zheng et al, 2020). In severe COVID-19 cases, elevated NF-kB activation

associated with impaired IFN-1 (Hadjadj *et al*, 2020) may be a host attempt to compensate for the lack of IFN-1 activation (Rubio *et al*, 2013), leading to NF-kB hyperactivation and release of pro-inflammatory cytokines. Also, SARS-CoV-1 viral papain-like proteases, contained within the nsp3 and nsp16 proteins, inhibit STING and its downstream IFN secretion (Chen *et al*, 2014). Perturbations in these pathways may impair the IFN response against SARS-CoV-2 and explain persistent blood viral load and an exacerbated inflammatory response in COVID-19 patients (Hadjadj *et al*, 2020).

*New crosstalk from interaction and text mining datasets*

New relationships emerging from associated interaction and text mining databases (see Section "Exploration of the networked knowledge" of the Results) suggested new pathway crosstalk (see Figs 4B and EV3). One of these was the interplay between ER stress and the immune pathways, as PPP1R15A regulates the expression of TNF and the translational inhibition of both IFN-1 and IL-6 (Smith, 2018). This finding coincided with the proposed interaction of pathways responsible for protein degradation and viral detection, as SQSTM1, an autophagy receptor and NFKB1 regulator, controls the activity of cGAS, a double-stranded DNA detector (Seo *et al*, 2018). Another association revealed by text mining data was ADAM17 and TNF release from the immune cells in response to ACE2-S protein interaction with SARS-CoV-1 (Haga *et al*, 2008), potentially increasing the risk of COVID-19 infection (Zipeto *et al*, 2020). This new interaction connected diagrams of the (i) "Viral replication cycle" via ACE2-S protein interactions, (ii) "Viral subversion of host defence mechanisms" via ER stress, (iii) "Host integrative stress response" via the renin–angiotensin system and (iv) "Host innate immune response" via pathways implicating TNF signalling.

*Novel regulators of protein activity*

Finally, we identified potential novel regulators of proteins in the C19DMap using interaction and text mining databases (see Fig 4C). These proteins take no part in the current version of the map but interact with molecules already represented in at least one of the diagrams. An example of such a novel regulator was NFE2L2, which controls the activity of HMOX1 in the context of viral infection (Kesic *et al*, 2011). In turn, HMOX1 controls immunomodulatory haem metabolism (Zhang *et al*, 2019), the mechanisms of viral replication, and is a target of SARS-CoV-2 Orf3a protein (Miao *et al*, 2020). The suggested NFE2L2-HMOX1 interaction is supported by the literature reports of NFE2L2 importance in COVID-19 cardiovascular complications due to crosstalk with the renin–angiotensin signalling pathway (Valencia *et al*, 2020) and potential interactions with viral entry mechanisms (Hassan *et al*, 2020). Interestingly, the modulation of the NFE2L2-HMOX1 axis was already proposed as a therapeutic measure for inflammatory diseases (Attucks *et al*, 2014), making it an appealing extension of the C19DMap.

**Computational analysis and modelling for hypothesis generation**

The standardised representation and programmatic access to the contents of the C19DMap support reproducible analytical and modelling workflows. Here, we discuss the range of possible

**Table 3.   Examples of computational workflows using the COVID-19 Disease Map**

| Workflow | COVID-19 Disease Map contents | User input | Tools | Output |
|---|---|---|---|---|
| Data interpretation | Online diagrams | Transcriptomics, Proteomics, Metabolomics | The MINERVA Platform (Gawron *et al*, 2016) PathVisio (Kutmon *et al*, 2015) Reactome (Jassal *et al*, 2020) | Visualisation of SARS-CoV-2 mechanisms contextualised to data[a] |
| | Diagrams in SIF format (via CasQ) | Transcriptomics | DoRothEA (Garcia-Alonso *et al*, 2019) | Contextualised evaluation of transcription factors activity under SARS-CoV-2 infection[b] |
| | Diagrams in SIF format (via CasQ) | Interactome data Proteomics | Network clustering (Messina *et al*, 2020) | Identification of new SARS-CoV-2-relevant interactions |
| Mechanistic modelling | Diagrams in SIF format (via CasQ) | TranscriptomicsMetabolomics | HiPATHIA (Salavert *et al*, 2016) CARNIVAL (Liu *et al*, 2019) | Endpoint prediction[c] Drug target effect prediction |
| Discrete modelling | Diagrams in SBML qual format (via CasQ) | Perturbation hypothesis (loss/gain of function) | CellCollective (Helikar *et al*, 2012) GINSim (Naldi *et al*, 2018a) BoolNet | Perturbation outcomes:[c] - Real-time simulation - Attractor analysis |
| Stochastic & Multiscale modelling | Diagrams in SBML qual format (via CasQ) | Perturbation hypothesis (loss/gain of function) | PhysiBoSS (Letort *et al*, 2019): MaBoSS (Stoll *et al*, 2017) + PhysiCell (Ghaffarizadeh *et al*, 2018) | Perturbation outcomes:[d] - Real-time simulation - Stochastic multiscale modelling |

Each workflow relies on the input from the C19DMap, either a direct diagram or its transformed contents, available in the GitLab repository. The workflow users may supply omics datasets to interpret them in the context of the map or test their hypotheses about how disease models will behave under specific perturbations.
[a]Example: see Results, Case study—analysis of cell-specific mechanisms using single-cell expression data.
[b]Example: see Results, Case study—RNA-Seq-based analysis of transcription factor activity.
[c]Example: see Results, Case study—RNA-Seq-based analysis of pathway signalling.
[d]https://colomoto.github.io/colomoto-docker/

approaches and demonstrate preliminary results, focusing on inter-operability, reproducibility and applicability of the methods and tools. Table 3 summarises selected computational workflows that can support data interpretation and hypothesis testing in COVID-19 research.

### Data interpretation and network analysis

The projection of omics data onto the C19DMap broadens and deepens our understanding of disease-specific mechanisms, in contrast to classical pathway enrichment analyses, which often produce lists of generic biological mechanisms. Visualisation of omics datasets on the map diagrams creates overlays, allowing interpretation of specific conditions, such as disease severity or cell types (Satagopam et al, 2016).

Datasets projected on the C19DMap can create signatures of molecular regulation determined by the expression levels of the corresponding molecules. Together, multiple omics readouts and multiple measurements can increase the robustness of such signatures (De Meulder et al, 2018). This interpretation can be extended using available SARS-CoV-2-related omics and interaction datasets (Bouhaddou et al, 2020) to infer which transcription factors, their target genes and signalling pathways are affected upon infection (Dugourd & Saez-Rodriguez, 2019). Combining regulatory interactions of the C19DMap with such data collections extends the scope of the analysis and may suggest new mechanisms to include in the map.

Besides the visual exploration of omics datasets, the network structure of the C19DMap allows extended network analysis of viral–human protein–protein interactions (PPIs) (Gordon et al, 2020). It can be expanded by merging virus–host with human PPIs and proteomics data to discover clusters of interactions indicating human biological processes affected by the virus (Messina et al, 2020). These clusters can be interpreted by visualising them on the C19DMap diagrams to reveal additional pathways or interactions to add to the map.

### Mechanistic and dynamic computational modelling

Diagrams from the C19DMap can be coupled with omics datasets to estimate their functional profiles and predict the effect of interventions, e.g. effects of drugs on their targets (Salavert et al, 2016). However, such an approach has a substantial computational complexity, limiting the size of the input diagrams. Large-scale mechanistic pathway modelling can address this challenge but requires transformation of diagrams into causal networks, which, combined with transcriptomics, (phospho-)proteomics or metabolomics data (Dugourd et al, 2021), contextualise the networks and hypotheses about intervention outcomes. Both approaches provide a set of coherent causal links connecting upstream drivers such as stimulations or pathogenic mutations to downstream changes in diagram endpoints or transcription factor activities.

Dynamic modelling allows analysis of changes of molecular networks in time to understand their complexity under disease-related perturbations (Naldi et al, 2018b). C19DMap diagrams, translated to SBML qual using CaSQ (see Materials and Methods), can be used in discrete modelling, using modelling software that supports SBML qual file import. Notably, multiscale processes involved in viral infection, from molecular interactions to multicellular behaviour, can be simulated using a dedicated computational architecture. In such a multiscale setup, single-cell models run in parallel to capture the behaviour of heterogeneous cell populations and their intercellular communications at different time scales, e.g. diffusion, cell mechanics, cell cycle, or signal transduction (Osborne et al, 2017; preprint: Wang et al, 2020). Implementing detailed COVID-19 signalling models in the PhysiBoSS framework (Letort et al, 2019) may help better understand complex dynamics of interactions between immune system components and the host cell.

### Case study: analysis of cell-specific mechanisms using single-cell expression data

To investigate cell-specific mechanisms of COVID-19, we projected single-cell expression data onto the C19DMap. To this end, we calculated differentially expressed genes (DEGs) for two datasets relevant to the disease. The first dataset describes non-infected bronchial secretory cells (Lukassen et al, 2020; Data ref: Lukassen et al, 2020), where we selected DEGs from three different subtypes of secretory cells dubbed (i) secretory1, (ii) secretory2 and (iii) secretory3 (transient) cells. The second dataset describes SARS-CoV-2-infected intestinal organoids (Triana et al, 2021; Data ref: Triana et al, 2021), where we selected DEGs from (iv) infected and (v) bystander immature enterocytes from the intestinal organoids infected with SARS-CoV-2. DEGs (i), (ii) and (iii) can serve as an illustration of pathway activity across normal lung cells, while datasets (iv) and (v) demonstrate a comparison of molecular activity in cells of infected intestinal tissue. These selected datasets are available as overlays in the C19DMap and can be interactively explored, showing cell-type-specific dysregulation of particular diagrams (see Materials and Methods).

Visual exploration of the differential expression profiles in the C19DMap revealed that transient secretory cells specifically express molecules associated with the virus replication cycle (TMPRSS2). This suggests that these cells are more susceptible to viral entry than the other types of bronchial secretory cells. Also, the interferon 1 signalling pathway was up-regulated in both secretory1 and transient secretory cells. However, transient secretory cells showed up-regulation of elements up- and downstream of the pathway (IFNAR1-JAK1, and ISG15 or OAS1); in secretory1 cells, the up-regulated proteins were downstream (transcription factor AP-1). In the intestinal organoid dataset, the comparison of infected and bystander immature enterocytes confirmed the downregulation of the ACE2 receptor reported by the original article (Triana et al, 2021; Data ref: Triana et al, 2021), as visualised in the virus replication cycle diagram. In addition, exploration of other affected pathways may suggest the context of this observation – for instance, the C19DMap demonstrated the differential activity of the pyrimidine deprivation pathway, which could suggest a reduction of transcriptional activity as a host response to the viral infection. Enrichment analysis of diagrams indicated that mitochondrial dysfunction, apoptosis, and inflammasome activation were dysregulated in infected enterocytes. The enrichment analysis of the cell-type-specific overlays was obtained by the GSEA plugin of the C19DMap. These results can be replicated and examined directly by the users via the visual interface of the C19DMap (see https://covid.pages.uni.lu/minerva-guide/ and Materials and Methods).

### Case study: RNA-Seq-based analysis of transcription factor activity

As discussed above, the diagrams of the C19DMap can be coupled with omics datasets. Here, we highlight how the map systematically

reveals the transcription factors (TFs) related to SARS-CoV-2 infection. To do so, we conducted differential expression analysis between SARS-CoV-2 infected Calu-3 human lung adenocarcinoma cell line and controls. Results were used to estimate TF activity deregulation upon viral infection. We mapped the outcomes of the TF activities to pathway diagrams of the C19DMap (see Materials and Methods).

The results for the interferon type I signalling diagram are shown in Fig 5. This pathway included some of the most active TFs after SARS-CoV-2 infection, such as STAT1, STAT2, IRF9 and NFKB1. These are well-known components of cytokine signalling and antiviral responses (Cheon *et al*, 2013; Fink & Grandvaux, 2013). Interestingly, these TFs were located downstream of various viral proteins (*E*, *S*, *Nsp1*, *Orf7a* and *Orf3a*) and members of the MAPK pathway (*MAPK8*, *MAPK14* and *MAP3K7*). SARS-CoV-2 infection is known to promote MAPK activation, which mediates the cellular response to pathogenic infection and promotes the production of pro-inflammatory cytokines (Bouhaddou *et al*, 2020). Overall, these results highlighted that the molecular mechanisms of the response of the human cells to SARS-CoV-2 infection can be investigated by combining omics datasets with the diagrams of the C19DMap.

### Case study: RNA-Seq-based analysis of pathway signalling

The diagrams of the C19DMap allow for a complex analysis of how the infection may affect signalling sequences in encoded pathways based on available omics data. To demonstrate this approach, we applied a mechanistic modelling algorithm that estimates the functional profiles of signalling circuits in the context of omics datasets. We used expression profiles from nasopharyngeal swabs of COVID-19 patients and controls (Lieberman *et al*, 2020; Data ref: Lieberman *et al*, 2020) to calculate the differential expression profiles and derive the pathway signalling activities (see Materials and Methods).

To illustrate this approach, we focused on the results of the analysis of the apoptosis pathway, also shown in Fig 6 and Table EV2. We observed an overall downregulation of both the CASP3 and CASP7 subpathways and an inhibition of the circuit ending in effector protein CASP3, possibly due to the downregulation of AKT1 and BAD and the downstream inhibition of BAX. Although the BAX downstream genes were up-regulated, the signal arriving at them was diminished by the effect of the previous nodes. Although CASP8 was up-regulated, the cumulative effect of the individual node activities resulted in the inhibition of CASP7. Indeed, inflammatory response via CASP8 has been described as a result of SARS-CoV-2

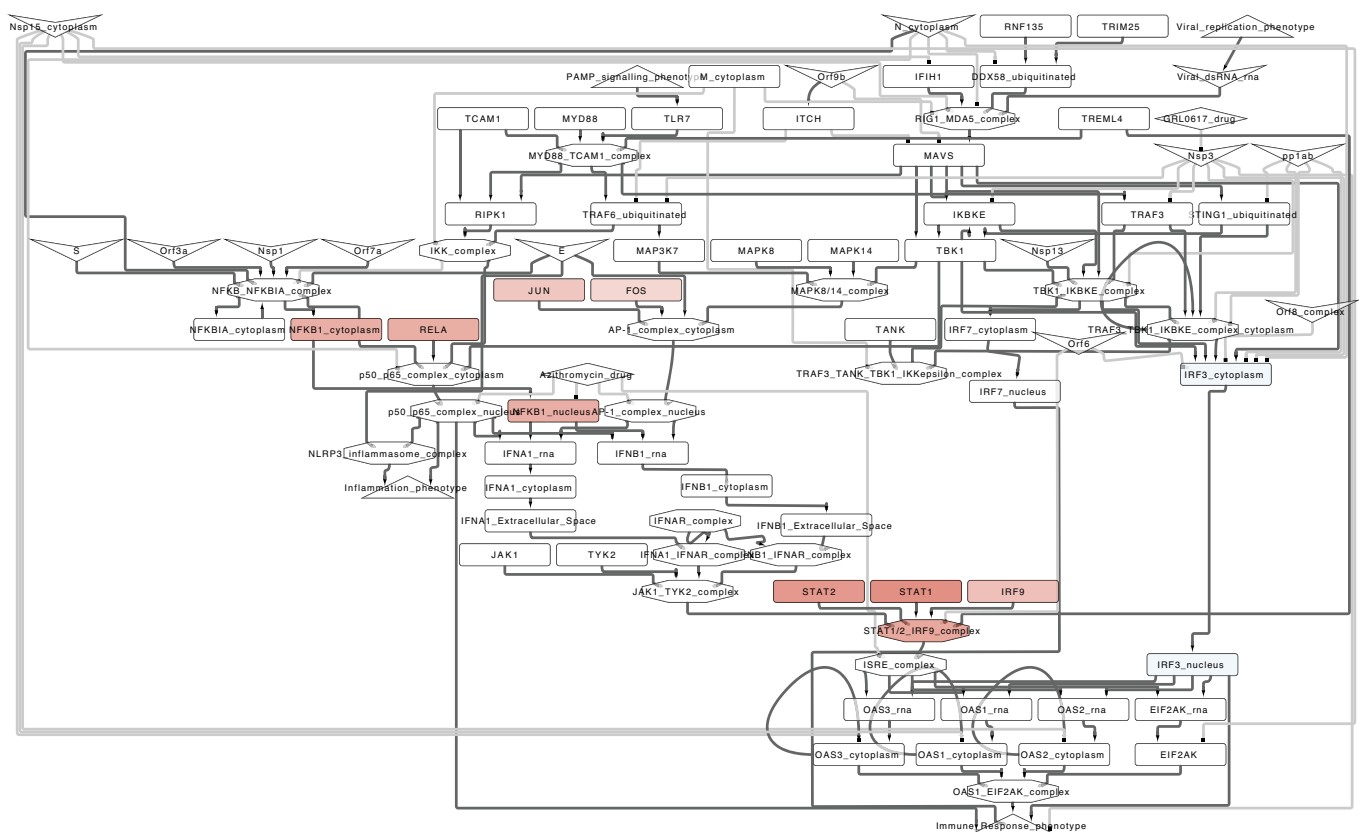

**Figure 5. *Interferon type I signalling pathway* diagram of the COVID-19 Disease Map integrated with TF activity derived from transcriptomics data after SARS-CoV-2 infection.**

A zoom was applied in the area containing the most active TFs (red nodes) after infection. Node shapes: host genes (rectangles), host molecular complex (octagons), viral proteins (V shape), drugs (diamonds) and phenotypes (triangles).

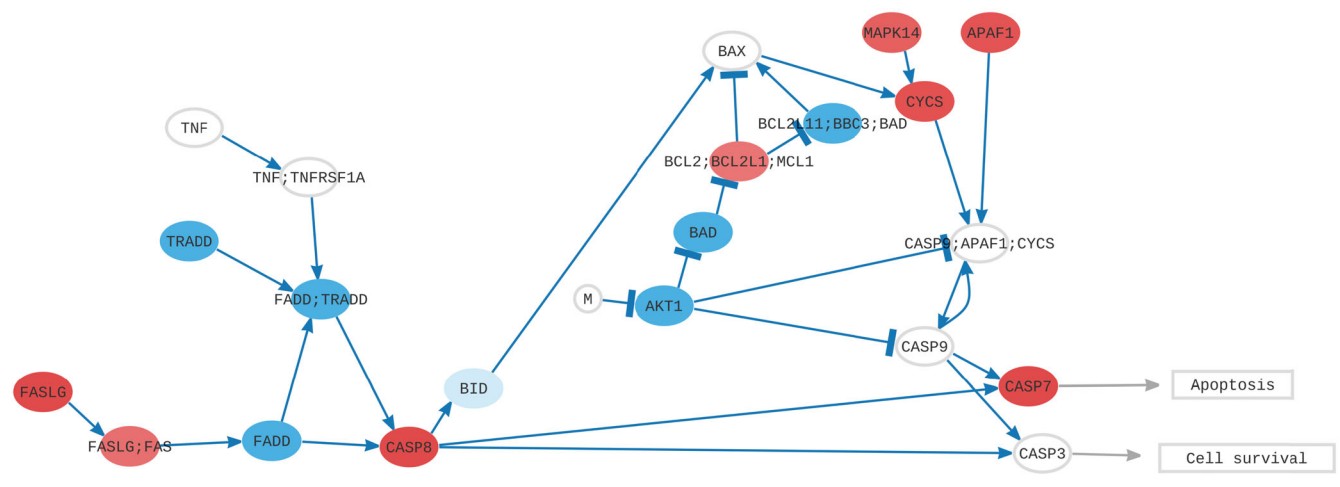

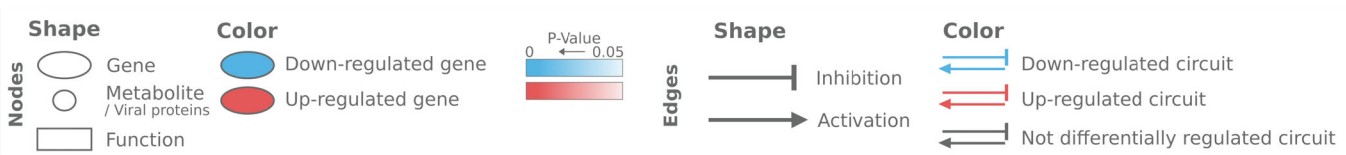

**Figure 6. Representation of the activation levels of apoptosis pathway in nasopharyngeal swabs from SARS-CoV-2-infected individuals.**

Activation levels were calculated using transcriptional data from GSE152075 and the Hipathia mechanistic pathway analysis algorithm. Each node represents a gene (ellipse), a human metabolite/viral protein (circle) or a function (rectangle). The pathway is composed of circuits from a receptor to an effector. Significant differential regulation of circuits in infected cells is highlighted by colour arrows (blue: inactive in infected cells). The colour of elements corresponds to the level of differential expression in SARS-CoV-2-infected human nasopharyngeal swabs versus non-infected nasopharyngeal human swabs. Blue: downregulated, red: up-regulated and white: no statistically significant differential expression.

infection (Li *et al*, 2020), and the role of caspase-induced apoptosis has been established, together with the ripoptosome/caspase-8 complex, as a pro-inflammatory checkpoint (Chauhan *et al*, 2018), which may be triggering up-regulation of such processes in other pathways. Overall, our findings recapitulate reported outcomes and provide explanations of the effects of interactions on pathway elements.

## Discussion

Our knowledge of COVID-19 molecular mechanisms is growing at a great speed, fuelled by global research efforts to investigate the pathophysiology of SARS-CoV-2 infection. Keeping an overview of all the findings, many of which focus on individual molecules, is a great challenge just one year after the start of the pandemic. The C19DMap aggregates this knowledge into molecular interaction diagrams, making it available for visual exploration by life science and clinical researchers and analysis by computational biologists.

The map complements and interfaces with other COVID-19 resources such as interaction databases (Licata *et al*, 2020; Perfetto *et al*, 2020), protein-centric resources (preprint: Lubin *et al*, 2020) and relevant omics data repositories (Delorey *et al*, 2021) by providing a context to particular pieces of information and helping with

data interpretation. The diagrams of the C19DMap describe molecular mechanisms of COVID-19, grounded in the relevant published SARS-CoV-2 research, completed where necessary by mechanisms discovered in related beta-coronaviruses.

We developed the contents of the C19DMap *de novo* in an unprecedented, community-driven effort involving independent biocurators, as well as WikiPathway and Reactome biocurators. Over forty diagrams with molecular resolution have been constructed since March 2020, shared across three platforms. In this work, we combined and harmonised expertise in biocuration across multiple teams, formulated clear guidelines and cross-reviewed the outcomes of our work with domain experts. Although the approach of community curation was applied in the past (Slayden *et al*, 2013; Naithani *et al*, 2019), we are not aware of any curation effort on a similar scale for a single human disease to date.

In this work, we established a computational framework accompanying the biocuration process, integrating interaction databases and text mining solutions to accelerate diagram building. This allowed us not only to enrich particular diagrams but also to explore crosstalk between them and prioritise key novel regulators of the encoded pathways. Thanks to the interoperability of different systems biology formats, we performed this analysis for diagrams constructed in different biocuration environments, extending current advances in pathway interoperability (Bohler *et al*, 2016).

Moreover, by developing reproducible analysis pipelines for the contents of the C19DMap, we promoted early harmonisation of formats, support of standards and transparency in all steps. Preliminary results of such efforts are illustrated in the case studies above. Notably, the biocurators and domain experts participated in the analysis helped to evaluate the outcomes and corrected the curated content if necessary. This way, we improve the quality of the analysis and increase the reliability of the models used to generate testable predictions.

The C19DMap is an open access repository of diagrams and reproducible workflows for content conversion and analysis. We followed FAIR principles in making our content and code available to the entire research community (Wilkinson *et al*, 2016). Importantly, FAIRDOMHub is an essential platform for disseminating all information about the project and linking contributors to their contributions. The C19DMap Community is open and expanding as more people with complementary expertise join forces. Using the FAIR approach for sharing the results of our work makes this effort more scalable. Recognising individual contributions and open access policy promote the distributed knowledge building and generation of research data.

The project aims to provide the tools to deepen our understanding of the mechanisms driving the infection and help boost drug development supported by testable suggestions. It offers insights into the dynamic nature of the disease at the molecular level and its propagation at the systemic level. Thus, it provides a platform for a precise formulation of models, accurate data interpretation, the potential for disease mitigation and drug repurposing. In the longer run, the constantly growing C19DMap content will be used to facilitate the finding of robust signatures related to SARS-CoV-2 infection predisposition, disease evolution or response to various treatments, along with the prioritisation of new potential drug targets or drug candidates.

This approach to an emerging worldwide pandemic leveraged the capacity and expertise of an entire swath of the bioinformatics community, bringing them together to improve the way we build and share knowledge. By aligning our efforts, we strive to provide COVID-19-specific pathway models, synchronise content with similar resources and encourage discussion and feedback at every stage of the curation process. Such an approach may help to deal with new waves of COVID-19 or similar pandemics in the long-term perspective.

# Materials and Methods

## Reagents and Tools table

| Reagent/Resource | Reference or Source | Identifier or Catalog Number |
|---|---|---|
| **Software** | | |
| CellDesigner v4.4.2 | http://www.celldesigner.org (Matsuoka *et al*, 2014) | |
| Newt v3.0 | https://newteditor.org | |
| SBGN-ED | Czauderna *et al* (2010) | |
| ySBGN | https://github.com/sbgn/ySBGN | |
| The MINERVA Platform v15.1.4 | https://minerva-web.lcsb.uni.lu (Gawron *et al*, 2016) | |
| Reactome | https://reactome.org (Jassal *et al*, 2020) | |
| WikiPathways | https://www.wikipathways.org (Slenter *et al*, 2018) | |
| PathVisio v3.3.0 | https://pathvisio.github.io (Kutmon *et al*, 2015) | |
| INDRA | Gyori *et al* (2017) | |
| AILANI COVID-19 | https://ailani.ai | |
| BioKB | https://biokb.lcsb.uni.lu/topic/DOID:0080599 | |
| OpenNLP + GNormPlus | https://opennlp.apache.org/ (Wei *et al*, 2015) | |
| COVIDminer | https://rupertoverall.net/covidminer | |
| rWikipathways v 1.12 | 10.18129/B9.bioc.rWikiPathways | |
| OmniPathR | https://github.com/saezlab/OmnipathR | |
| The MINERVA Conversion API v15.1 | https://minerva.pages.uni.lu/doc/api/15.1/converter/ (Hoksza *et al*, 2020) | |
| cd2sbgml | https://github.com/sbgn/cd2sbgnml (Balaur *et al*, 2020) | |
| rnef2sbgn | https://github.com/golovatenkop/rnef2sbgn | |
| Seurat v4.0 | https://satijalab.org/seurat/ (Hao *et al*, 2021) | |
| DESeq2 | 10.18129/B9.bioc.DESeq2 (Love *et al*, 2014) | |
| Viper v1.26.0 | 10.18129/B9.bioc.viper (Alvarez *et al*, 2016) | |
| DoRothEA v1.4.1 | 10.18129/B9.bioc.dorothea (Garcia-Alonso *et al*, 2019) | |
| CaSQ v0.9.11 | Aghamiri *et al* (2020) | |

**Reagents and Tools table**   (continued)

| Reagent/Resource | Reference or Source | Identifier or Catalog Number |
|---|---|---|
| CoV-HiPathia | Rian *et al* (2021) | |
| **Datasets** | | |
| IMEx Consortium COVID-19 dataset | Perfetto *et al* (2020) | |
| SIGNOR 2.0 COVID-19 dataset | Licata *et al* (2020) | |
| OmniPath | Türei *et al* (2021) | |
| INDRA EMMAA Collection, accessed: 2020.12.01 | https://emmaa.indra.bio/dashboard/covid19 | |
| RNA-Seq transcriptomic single-cell profiles | https://eils-lung.cells.ucsc.edu https://doi.org/10.6084/m9.figshare.11981034.v1 (Lukassen *et al*, 2020) | |
| SARS-CoV-2-infected intestinal organoids | Triana *et al* (2021) | GSE156760 |
| SARS-CoV-2-infected Calu-3 cells | Blanco-Melo *et al* (2020) | GSE147507 |
| SARS-CoV-2 nasopharyngeal swabs | Lieberman *et al* (2020) | GSE152075 |

## Methods and Protocols

### Biocuration platforms

Individual diagrams were encoded in systems biology layout-aware formats (see below) by biocurators using CellDesigner (Matsuoka *et al*, 2014), Newt (https://newteditor.org), SBGN-ED (Czauderna *et al*, 2010) and ySBGN (https://github.com/sbgn/ySBGN). This community-based curation was coordinated by sharing curation topics, e.g. relevant pathways or particular SARS-CoV-2 proteins across the community to cover the available literature and identify synergies. Curation guidelines (https://fairdomhub.org/documents/661) were established to ensure proper representation and annotation of the key features of the diagrams. Curation guidelines for logical models (Niarakis *et al*, 2020) were followed. Regular technical reviews of the diagrams were performed following a previously established checklist to harmonise their content. The diagrams are stored and versioned in a GitLab repository (https://gitlab.lcsb.uni.lu/covid/models). Individual diagrams are visualised in the MINERVA Platform (Gawron *et al*, 2016). The entry-level view is based on Fig 2.

Reactome (Jassal *et al*, 2020) biocuration efforts initially focused on SARS-CoV-1 and its proteins, and their functions are extensively documented in the experimental literature. Reactome curators were assigned a subpathway from the viral life cycle, a host pathway or potential therapeutics. Curators were supported by an editorial manager and a dedicated SARS literature triage process. The resulting set of pathways for SARS-CoV-1 provided the basis for computational inference of the corresponding SARS-CoV-2 pathways based on structural and functional homologies between the two viruses. The computationally inferred SARS-CoV-2 infection pathway events and entities were then reviewed and manually curated using published SARS-CoV-2 experimental data. Reactome diagrams are available via a dedicated pathway collection (https://reactome.org/PathwayBrowser/#/R-HSA-9679506).

The WikiPathways (Slenter *et al*, 2018) diagrams were constructed using PathVisio (Kutmon *et al*, 2015), with annotation of pathway elements from the integrated BridgeDb identifier mapping framework (van Iersel *et al*, 2010). All pathways are stored in GPML format (Kutmon *et al*, 2015). The WikiPathways diagrams are available via a dedicated pathway portal, grouping pathway models specific to SARS-CoV-2, other coronaviruses and general cellular processes relevant to the virus–host interactions (https://www.wikipathways.org/index.php/Portal:COVID-19).

### Layout-aware systems biology formats

The diagrams are available in SBML format (Keating *et al*, 2020), allowing computational modelling of biological processes. SBML stores visual information about encoded elements and reactions using render (Bergmann *et al*, 2018) and layout (Gauges *et al*, 2015) packages. An early version of SBML adapted by CellDesigner allows storing layout and rendering information. Systems Biology Graphical Notation (SBGN) format is a graphical standard for visual encodings of molecular entities and their interactions, implemented using SBGNML (Bergmann *et al*, 2020) for encoding the layout of SBGN maps and their annotations. Finally, GPML (Kutmon *et al*, 2015) is a structured XML format for computable representation of biological knowledge used by the WikiPathways platform.

Interactions and interacting entities are annotated following a uniform, persistent identification scheme, using either MIRIAM Registry or Identifiers.org (Juty *et al*, 2012) and the guidelines for annotations of computational models. Viral protein interactions are explicitly annotated with their taxonomy identifiers to highlight findings from strains other than SARS-CoV-2. Stable protein complexes from SARS-CoV-2 and SARS are annotated using the Complex Portal.

### Interaction databases

The biocuration process was supported by interaction and pathway databases storing structured, annotated and curated information about COVID-19 virus–host interactions. The IMEx Consortium (Meldal *et al*, 2019) dataset (Perfetto *et al*, 2020) contains curated Coronaviridae-related interaction data from reviewed manuscripts and preprints, resulting in a dataset of roughly 7,300 interactions extracted from over 250 publications, including data from SARS-CoV-2, SARS, CoV, and other strains of Coronaviridae. The dataset is updated with every release of IMEx data and is open access (https://www.ebi.ac.uk/intact/resources/datasets#coronavirus). The SIGNOR 2.0 (Licata *et al*, 2020) dataset contains manually annotated and validated signalling interactions related to the host–virus interaction, including cellular pathways modulated during SARS-CoV-2 infection. The dataset was constructed from the literature on causal interactions between SARS-CoV-2, SARS-COV-1, MERS proteins and the human host and is openly available (https://signor.

uniroma2.it/covid/). The Elsevier Pathway Collection (Daraselia et al, 2004; Nesterova et al, 2020) COVID-19 dataset comprises manually reconstructed and annotated pathway diagrams. Statements about molecular interactions are extracted into a knowledge graph by a dedicated text mining technology adapted for extracting facts about viral proteins and viruses from the literature. These interactions were filtered for experimental evidence, used for pathway reconstruction and made openly available (http://dx.doi.org/10.17632/d55xn2c8mw.1). Information from OmniPath (Türei et al, 2021) on existing interactions gathered from pathway and interaction databases was used in a programmatic way to suggest cell-specific interactions and cell–cell interactions specific to immune reactions.

### Text and figure mining

Text mining was performed on the CORD-19: COVID-19 Open Research Dataset dataset (preprint: Lu Wang et al, 2020). INDRA (Gyori et al, 2017), AILANI COVID-19 (https://ailani.ai) and BioKB processed CORD-19 dataset (https://biokb.lcsb.uni.lu/topic/DOID:0080599), with their results available programmatically via REST API and SPARQL interfaces. An OpenNLP-based (https://opennlp.apache.org/) text mining workflow using GNormPlus (Wei et al, 2015) was applied to the CORD-19 dataset and the collection of MEDLINE abstracts associated with the genes in the SARS-CoV-2 PPI network (Gordon et al, 2020) using the Entrez GeneRIFs, https://www.ncbi.nlm.nih.gov/gene/about-generif. (https://gitlab.lcsb.uni.lu/covid/models/-/tree/master/Resources/Text%20mining). Also, we used data from 221 CORD-19 dataset figures using a dedicated Figure Mining Workflow (Hanspers et al, 2020), with results available at https://gladstone-bioinformatics.shinyapps.io/shiny-covidpathways. Results of text mining were accessed by the curators in the form of molecular interactions with references to the articles and to sentences from which these interactions were derived. We systematically aligned the C19DMap with assembled INDRA Statements, both to enrich and to extend the map (see "Crosstalk analysis" below). The content of INDRA and AILANI COVID-19 was accessible via interfaces that allow users to provide natural language queries, such as "What are COVID-19 risk factors?" or "What are the interactors of ACE2?", facilitating extracting knowledge from the results of text mining workflows. The results of the INDRA workflow were visualised using the COVIDminer project (https://rupertoverall.net/covidminer). Each extracted statement describes a directed interaction between two gene products, small molecules or biological processes. The causal network representing the COVIDminer database is browsable through a web interface. The results of the OpenNLP-based text mining workflow were imported into a BioKC biocuration platform for structured processing and SBML export.

### Crosstalk analysis

Crosstalk analysis was performed for the list of C19DMap diagrams (Table EV1). The code is available at: https://gitlab.lcsb.uni.lu/covid/models/-/tree/master/Resources/Crosstalks. Individual diagrams were accessed via the API of the MINERVA Platform, WikiPathway diagrams via the rWikipathways package (https://github.com/wikipathways/rWikiPathways) and Reactome diagrams via the Reactome API. Text mining interactions are from the INDRA EMMAA Collection (https://emmaa.indra.bio/dashboard/covid19), dataset timestamp: 2020-12-01-21-05-54. Verified molecular interactions for

quality control of the text mining data were obtained from OmniPath using the OmnipathR package (https://github.com/saezlab/OmnipathR). We filtered text mining interactions of the EMMAA dataset for "belief" of 0.8 or higher and retained those matching the direction and interacting molecules to the OmniPath dataset. We call this filtered group of interactions "EMMAA-OP interactions".

Crosstalk between C19DMap diagrams was calculated based on the HGNC identifiers of their elements. For simplification, all elements of the same diagram were considered to be interacting with each other. Three types of networks were constructed: existing crosstalk, new crosstalk and new regulators. Diagram groups followed the scheme in the list of C19DMap diagrams (Table EV1). The networks were visualised using Cytoscape (Shannon et al, 2003). The colour code is common for the networks: light green for nodes representing a diagram or a diagram group, light blue for nodes having one or two neighbours, yellow for nodes having three or four neighbours and red for nodes with five or more neighbours. Diagram nodes have prefixes indicating their provenance. Diagram groups have no prefixes, as they combine diagrams across platforms. Existing crosstalk between diagrams, or groups of diagrams, was calculated by identifying shared HGNC identifiers linking diagrams or groups of diagrams. To calculate new crosstalk between diagrams, we merged the EMMAA-OP interactions with the network of existing crosstalk and kept only those new interactions that link at least two upstream and two downstream diagrams or diagram groups. To calculate new upstream regulators of existing diagrams, we merged the EMMAA-OP interactions with the network of existing crosstalk. We kept interactions with source elements, not within existing diagrams, and target elements in at least one existing diagram or diagram group.

### Diagram interoperability and translation for computational modelling

Bidirectional translation of curated diagrams between CellDesigner and SBGNML formats is supported by the MINERVA Conversion API https://minerva.pages.uni.lu/doc/api/15.1/converter/ (Hoksza et al, 2020), and cd2sbgml converter https://github.com/sbgn/cd2sbgnml (Balaur et al, 2020). The MINERVA Conversion API supports bidirectional translation between CellDesigner, SBML and GPML. Unidirectional translation from Reactome format to GPML is supported by the Reactome-to-WikiPathways converter (Bohler et al, 2016). Diagrams in the RNEF format of Elsevier Pathway Studio were translated to SBGNML using a dedicated rnef2sbgn software (https://github.com/golovatenkop/rnef2sbgn).

The C19DMap diagrams (Table EV1) in CellDesigner format were translated using CaSQ (Aghamiri et al, 2020) into executable Boolean networks. Conversion rules and logical formulae were inferred according to the topology and the annotations of the diagrams. SBML-qual files (Chaouiya et al, 2013) generated with CaSQ (Aghamiri et al, 2020) retained their references, annotations and layout of the original CellDesigner file. They can be used for *in silico* simulations and analysis with CellCollective (Helikar et al, 2012), GINsim (Naldi et al, 2018a) or MaBoSS (Stoll et al, 2017). CaSQ was adapted to produce SIF files necessary for HiPATHIA (Hidalgo et al, 2017) and CARNIVAL (Liu et al, 2019) pipelines. The C19DMap GitLab repository (https://gitlab.lcsb.uni.lu/covid/models) was configured to translate stable versions of diagrams into SBML qual and SIF files. The diagrams were translated to XGMML using Cytoscape and GINSim.

### Calculation and visualisation of single-cell RNA-Seq expression profiles

RNA-Seq transcriptomic single-cell profiles were calculated for (i) non-infected airway cells (Lukassen *et al*, 2020; Data ref: Lukassen *et al*, 2020) and (ii) SARS-CoV-2-infected intestinal organoids (Triana *et al*, 2021; Data ref: Triana *et al*, 2021). The Seurat package (Hao *et al*, 2021) was used to calculate cell-specific transcriptional profiles. For dataset (i), differential expression was calculated using every cell type against remaining cell types and applying the *FindAllMarkers* function of the Seurat package with min pct 0.25 and log fold change threshold 0.25. For dataset (ii), the cells were classified into bystander or infected based on the absence or presence of SARS-CoV-2 mRNA measured by scRNAseq (Triana *et al*, 2021; Data ref: Triana *et al*, 2021). Differential expression was calculated by contrasting the mock organoids with the bystander or infected cells after 12 h or 24 h of treatment. Expression profiles for the following cell types and conditions were selected for visualisation and enrichment analysis: for dataset (i), three types of secretory cells; and for dataset (ii), infected and bystander immature enterocytes 24 h post-infection versus mock. The datasets were selected to recapitulate the findings in the original papers and demonstrate the capability of the C19DMap for cell-specific data interpretation. Selected differentially expressed genes (DEGs) were prepared for visualisation in the MINERVA Platform as follows. Differential expression values were normalised to $[-1,1]$ range by dividing by three and setting the outliers to their respective border values. Expression values and their corresponding HGNC symbols were used to create visual overlays in the C19DMap in the MINERVA Platform (https://covid19map.elixir-luxembourg.org/minerva/index.xhtml?id=covid19_map_17Jun21, "Overlays" tab). On-the-fly exploration and enrichment analyses using the GSEA plugin (Hoksza *et al*, 2019) are described in a dedicated guide (https://covid.pages.uni.lu/minerva-guide/). Complete expression analysis and transformation scripts are available in RMarkdown files at https://gitlab.lcsb.uni.lu/covid/models/-/tree/master/Resources/Omics%20analysis.

### RNA-Seq-based analysis of transcription factor activity

RNA-Seq transcriptomic profiles of SARS-CoV-2 infection come from SARS-CoV-2-infected Calu-3 cells measured 24 h after infection, Gene Expression Omnibus reference GSE147507 (Blanco-Melo *et al*, 2020; Data ref: Blanco-Melo *et al*, 2020). Differential expression analysis of the transcript abundances between conditions was performed with DESeq2 (Love *et al*, 2014). The resulting *t*-values from the differential expression analysis were used to estimate the effect of SARS-CoV-2 at the transcription factor (TF) activity level. This analysis was performed using the software Viper (Alvarez *et al*, 2016) algorithm coupled with TF–target interactions from DoRothEA (Garcia-Alonso *et al*, 2019). DoRothEA TF–target interactions have a confidence level based on the reliability of their source, which ranges from A (most reliable) to E (least reliable). Here, interactions with confidence levels A, B and C were selected. Activities of TFs having at least five different targets were computed. The TFs normalised enrichment score from the Viper output was mapped on the "Interferon type I signalling pathway diagram" (https://fairdomhub.org/models/713) of the C19DMap using the SIF files generated by CaSQ. The resulting network was visualised using Cytoscape (Shannon *et al*, 2003). Notebooks to reproduce the results of this case study are available at https://github.com/saezlab/Covid19.

### RNA-seq-based analysis of pathway signalling

The CoV-HiPathia (Rian *et al*, 2021) web tool was used to calculate the level of activity of the subpathways of the apoptosis diagram (https://fairdomhub.org/models/712) from the C19DMap. RNA-Seq transcriptomic profiles come from a public dataset of nasopharyngeal swabs from 430 individuals with SARS-CoV-2 and 54 negative controls, Gene Expression Omnibus reference GSE152075 (Lieberman *et al*, 2020; Data ref: Lieberman *et al*, 2020). RNA-Seq gene expression data with the trimmed mean of M-values (TMM) normalisation (Robinson *et al*, 2010) were rescaled to the range [0;1] for the calculation of the signal and normalised using quantile normalisation (Bolstad *et al*, 2003). Normalised gene expression values and the experimental design (case/control sample names files) were uploaded to CoV-Hipathia to calculate the level of activation of the signalling in the selected diagram. A case/control contrast with a Wilcoxon test was used to assess differences in signalling activity between the two conditions. To reproduce the results, files with normalised gene expression data and the experimental design can be generated using the code https://gitlab.lcsb.uni.lu/covid/models/-/tree/master/Resources/Hipathia/data_preprocessing. These files can then be used in CoV-HiPathia at http://hipathia.babelomics.org/covid19/ under the "Differential signalling" tab. Diagrams from the C19DMap can be selected in the "Pathway source" section, under "Disease Maps Community curated pathways".

## Data availability

COVID-19 Disease Map diagrams are available via:
-the GitLab repository (https://gitlab.lcsb.uni.lu/covid/models).
-WikiPathways collection (http://covid.wikipathways.org).
-Reactome collection (https://reactome.org/PathwayBrowser/#/R-HSA-9679506).

Workflows, executable models and network models are available via:
-the GitLab repository (https://gitlab.lcsb.uni.lu/covid/models).
-FAIRDOMHub (https://fairdomhub.org/projects/190).

**Expanded View** for this article is available online.

## Acknowledgements

We would like to thank Andjela Tatarovic, architect, and Gina Crovetto, a researcher in the field of cancer, for their help with the design of the top-level view diagrams. We would like to acknowledge the Responsible and Reproducible Research (R3) team of the Luxembourg Centre for Systems Biomedicine for supporting the project and providing necessary communication and data sharing resources. The work presented in this paper was carried out using the ELIXIR Luxembourg tools and services. This study was supported by the Luxembourg National Research Fund (FNR) COVID-19 Fast-Track grant programme, grant COVID-19/2020-1/14715687/CovScreen (E. Glaab); European Commission, INFORE grant H2020-ICT-825070 (A. Montagud, M. Ponce de Leon, M. Vazques and A. Valencia); European Commission, PerMedCoE grant H2020-ICT-951773 (A. Montagud, M. Ponce de Leon, M. Vazques and A. Valencia) the Federal Ministry of Education and Research (BMBF, Germany) and the Baden-Württemberg Ministry of Science, the Excellence Strategy of the German Federal and State

Governments (A. Renz); German Center for Infection Research (DZIF), grant no 8020708703 (A. Dräger); The Netherlands Organisation for Health Research and Development (ZonMw), grant no 10430012010015, (M. Kutmon, S. Coort, F. Ehrhart, N. Pham, E.L. Willighagen, C.T. Evelo); H2020 Marie Skłodowska-Curie Actions, grant number 765274 (J. Scheel); National Institutes of Health, USA (NIH), grant number U41 HG003751 (L.D. Stein). The development of Reactome is supported by grants from the US National Institutes of Health (U41 HG003751) and the European Molecular Biology Laboratory.

## Author contributions

MO, AN, AM and IK planned and coordinated the project. RP, AO-R, J-MR, RF, VO and SM advised the project as domain experts. MO, AN, AM, IK, VS, SSA, MLA, EG, AR, GF, CM, BB, GF, LCMG, JS, MH, SG, JS, HB, TC, FS, AM, MPL, AF, YH, NH, TGY, AD, AR, MN, ZB, FM, DB, LF, MC, MR, VN, JV, LS, MW, EEA, JS, JZ, KO, JT, EK, GYS, KH, MK, SC, LE, FE, DABR, DS, MM, NP, RH, BJ, LM, MO-M, ASR, KR, VS, RS, CS and TV curated and reviewed the diagrams. MO, PG, ES, LH, VS, GW, AR, MG, SO, CG and XH designed, developed and implemented key elements of the data sharing and communication infrastructure. RW. Overall, DM, AB, BMG, JAB, CV, VG, MV, PP, LL, MI, FS, AN, AY and AW designed and developed the contents of interaction and pathway databases, and text mining platforms and their visualisation and interoperability functionalities. AN, DT, AL, OB, SS, AV, ME, MP, KR, TH, BLP, DM, AT, MO, BDM, SB, AD, AN, VN and LC developed format interoperability, analysis and modelling workflows. CS, ED, TK, TF, FA, JSB, JH, OW, ELW, ARP, CTE, MEG, LS, HH, PD'E, JS-R, JD, AV, HK, EB, CA, RB and RS defined the strategy and scope of the project and revised its progress. MO, AN, AM and IK wrote the manuscript. AO-R, IK and AM designed the overview figures. AW, PD'E, JSB and LDS revised and contributed significantly to the structure of the manuscript. All authors have revised, read and accepted the manuscript in its final form.

## Conflict of interest

A. Niarakis collaborates with SANOFI-AVENTIS R&D via a public–private partnership grant (CIFRE contract, n° 2020/0766). D. Maier and A. Bauch are employed at Biomax Informatics AG and will be affected by any effect of this publication on the commercial version of the AILANI software. J.A. Bachman and B. Gyori received consulting fees from Two Six Labs, LLC. T. Helikar has served as a shareholder and/or has consulted for Discovery Collective, Inc. R. Balling and R. Schneider are founders and shareholders of MEGENO S.A. and ITTM S.A. J. Saez-Rodriguez receives funding from GSK and Sanofi and consultant fees from Travere Therapeutics. The remaining authors have declared that they have no Conflict of interest.

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

---

## List of affiliations

Marek Ostaszewski[1,*]; Anna Niarakis[2,3]; Alexander Mazein[1]; Inna Kuperstein[4,5,6]; Robert Phair[7]; Aurelio Orta-Resendiz[8,9]; Vidisha Singh[2]; Sara Sadat Aghamiri[10]; Marcio Luis Acencio[1]; Enrico Glaab[1]; Andreas Ruepp[11]; Gisela Fobo[11]; Corinna Montrone[11]; Barbara Brauner[11]; Goar Frishman[11]; Luis Cristóbal Monraz Gómez[4,5,6]; Julia Somers[12]; Matti Hoch[13]; Shailendra Kumar Gupta[13]; Julia Scheel[13]; Hanna Borlinghaus[14]; Tobias Czauderna[15]; Falk Schreiber[14,15]; Arnau Montagud[16]; Miguel Ponce de Leon[16]; Akira Funahashi[17]; Yusuke Hiki[17]; Noriko Hiroi[18]; Takahiro G Yamada[17]; Andreas Dräger[19,20,21]; Alina Renz[19,20]; Muhammad Naveez[13,22]; Zsolt Bocskei[23]; Francesco Messina[24,25]; Daniela Börnigen[26]; Liam Fergusson[27]; Marta Conti[28]; Marius Rameil[28]; Vanessa Nakonecnij[28]; Jakob Vanhoefer[28]; Leonard Schmiester[28,29]; Muying Wang[30]; Emily E Ackerman[30]; Jason E Shoemaker[30,31]; Jeremy Zucker[32]; Kristie Oxford[32]; Jeremy Teuton[32]; Ebru Kocakaya[33]; Gökçe Yağmur Summak[33]; Kristina Hanspers[34]; Martina Kutmon[35,36]; Susan Coort[35]; Lars Eijssen[35,37]; Friederike Ehrhart[35,37]; Devasahayam Arokia Balaya Rex[38]; Denise Slenter[35]; Marvin Martens[35]; Nhung Pham[35]; Robin Haw[39]; Bijay Jassal[39]; Lisa Matthews[40]; Marija Orlic-Milacic[39]; Andrea Senff Ribeiro[39,41]; Karen Rothfels[39]; Veronica Shamovsky[40]; Ralf Stephan[39]; Cristoffer Sevilla[42]; Thawfeek Varusai[42]; Jean-Marie Ravel[43,44]; Rupsha Fraser[45]; Vera Ortseifen[46]; Silvia Marchesi[47]; Piotr Gawron[1,48]; Ewa Smula[1]; Laurent Heirendt[1]; Venkata Satagopam[1]; Guanming Wu[49]; Anders Riutta[34]; Martin Golebiewski[50]; Stuart Owen[51]; Carole Goble[51]; Xiaoming Hu[50]; Rupert W Overall[52,53,54]; Dieter Maier[55]; Angela Bauch[55]; Benjamin M Gyori[56]; John A Bachman[56]; Carlos Vega[1]; Valentin Grouès[1]; Miguel Vazquez[16]; Pablo Porras[42]; Luana Licata[57]; Marta Iannuccelli[57]; Francesca Sacco[57]; Anastasia Nesterova[58]; Anton Yuryev[58]; Anita de Waard[59]; Denes Turei[60]; Augustin Luna[61,62]; Ozgun Babur[63]; Sylvain Soliman[3]; Alberto Valdeolivas[60]; Marina Esteban-Medina[64,65]; Maria Peña-Chilet[64,65,66]; Kinza Rian[64,65]; Tomáš Helikar[3]; Bhanwar Lal Puniya[67]; Dezso Modos[68,69]; Agatha Treveil[68,69]; Marton Olbei[68,69]; Bertrand De Meulder[70]; Stephane Ballereau[71]; Aurélien Dugourd[60,72]; Aurélien Naldi[3]; Vincent Noël[4,5,6]; Laurence Calzone[4,5,6]; Chris Sander[61,62]; Emek Demir[12]; Tamas Korcsmaros[68,69]; Tom C Freeman[73]; Franck Augé[23]; Jacques S Beckmann[74]; Jan Hasenauer[75,76]; Olaf Wolkenhauer[13]; Egon L Willighagen[35]; Alexander R Pico[34]; Chris T Evelo[35,36]; Marc E Gillespie[39,77]; Lincoln D Stein[39,78]; Henning Hermjakob[42]; Peter D'Eustachio[40]; Julio Saez-Rodriguez[60]; Joaquin Dopazo[64,65,66,79]; Alfonso Valencia[16,80]; Hiroaki Kitano[81,82]; Emmanuel Barillot[4,5,6]; Charles Auffray[71]; Rudi Balling[1]; Reinhard Schneider[1]; the COVID-19 Disease Map Community[†]

---

[1]Luxembourg Centre for Systems Biomedicine, University of Luxembourg, Esch-sur-Alzette, Luxembourg. [2]Université Paris-Saclay, Laboratoire Européen de Recherche pour la Polyarthrite rhumatoïde - Genhotel, Univ Evry, Evry, France. [3]Lifeware Group, Inria Saclay-Ile de France, Palaiseau, France. [4]Institut Curie, PSL Research University, Paris, France. [5]INSERM, Paris, France. [6]MINES ParisTech, PSL Research University, Paris, France. [7]Integrative Bioinformatics, Inc., Mountain View, CA, USA. [8]Institut Pasteur, Université de Paris, Unité HIV, Inflammation et Persistance, Paris, France. [9]Bio Sorbonne Paris Cité, Université de Paris, Paris, France. [10]Inserm- Institut national de la santé et de la recherche médicale, Paris, France. [11]Institute of Experimental Genetics (IEG), Helmholtz Zentrum München-German Research Center for Environmental Health (GmbH), Neuherberg, Germany. [12]Department of Molecular and Medical Genetics, Oregon Health & Sciences University, Portland, OR, USA. [13]Department of Systems Biology and Bioinformatics, University of Rostock, Rostock, Germany. [14]Department of Computer and Information Science, University of Konstanz, Konstanz, Germany. [15]Faculty of Information Technology, Department of Human-Centred Computing, Monash University, Clayton, Vic., Australia. [16]Barcelona Supercomputing Center (BSC), Barcelona, Spain. [17]Department of Biosciences and Informatics, Keio University, Yokohama, Japan. [18]Graduate School of Media and Governance, Research Institute at SFC, Keio University, Kanagawa, Japan. [19]Computational Systems Biology of Infections and Antimicrobial-Resistant Pathogens, Institute for Bioinformatics and Medical Informatics (IBMI), University of Tübingen, Tübingen, Germany. [20]Department of Computer Science, University of Tübingen, Tübingen, Germany. [21]German Center for Infection Research (DZIF), partner site, Tübingen, Germany. [22]Institute of Applied Computer Systems, Riga Technical University, Riga, Latvia. [23]Sanofi R&D, Translational Sciences, Chilly-Mazarin, France. [24]Dipartimento di Epidemiologia Ricerca Pre-Clinica e Diagnostica Avanzata, National Institute for Infectious Diseases 'Lazzaro Spallanzani' I.R.C.C.S., Rome, Italy. [25]COVID-19 INMI Network Medicine for IDs Study Group, National Institute for Infectious Diseases 'Lazzaro Spallanzani' I.R.C.C.S, Rome, Italy. [26]Bioinformatics Core Facility, Universitätsklinikum Hamburg-Eppendorf, Hamburg, Germany. [27]Royal (Dick) School of Veterinary Medicine, The University of Edinburgh, Edinburgh, UK. [28]Faculty of Mathematics and Natural Sciences, University of Bonn, Bonn, Germany. [29]Center for Mathematics, Chair of Mathematical Modeling of Biological Systems, Technische Universität München, Garching, Germany. [30]Department of Chemical and Petroleum Engineering, University of Pittsburgh, Pittsburgh, PA, USA. [31]Department of Computational and Systems Biology, University of Pittsburgh, Pittsburgh, PA, USA. [32]Pacific Northwest National Laboratory, Richland, WA, USA. [33]Stem Cell Institute, Ankara University, Ankara, Turkey. [34]Institute of Data Science and Biotechnology, Gladstone Institutes, San Francisco, CA, USA. [35]Department of Bioinformatics - BiGCaT, NUTRIM, Maastricht University, Maastricht, The Netherlands. [36]Maastricht Centre for Systems Biology (MaCSBio), Maastricht University, Maastricht, The Netherlands. [37]Maastricht University Medical Centre, Maastricht, The Netherlands. [38]Center for Systems Biology and Molecular Medicine, Yenepoya (Deemed to be University), Mangalore, India. [39]MaRS Centre, Ontario Institute for Cancer Research, Toronto, ON, Canada. [40]NYU Grossman School of Medicine, New York, NY, USA. [41]Universidade Federal do Paraná, Curitiba, Brasil. [42]European Bioinformatics Institute (EMBL-EBI), European Molecular Biology Laboratory, Hinxton, Cambridgeshire, UK. [43]INSERM UMR_S 1256, Nutrition, Genetics, and Environmental Risk Exposure (NGERE), Faculty of Medicine of Nancy, University of Lorraine, Nancy, France. [44]Laboratoire de génétique médicale, CHRU Nancy, Nancy, France. [45]Queen's Medical Research Institute, The University of Edinburgh, Edinburgh, UK. [46]Senior Research Group in Genome Research of Industrial Microorganisms, Center for Biotechnology, Bielefeld University, Bielefeld, Germany. [47]Department of Surgical Science, Uppsala University, Uppsala, Sweden. [48]Institute of Computing Science, Poznan University of Technology, Poznan, Poland. [49]Department of Medical Informatics and Clinical Epidemiology, Oregon Health & Science University, Portland, OR, USA. [50]Heidelberg Institute for Theoretical Studies (HITS), Heidelberg, Germany. [51]Department of Computer Science, The University of Manchester, Manchester, UK. [52]German Center for Neurodegenerative Diseases (DZNE) Dresden, Dresden, Germany. [53]Center for Regenerative Therapies Dresden (CRTD), Technische Universität Dresden, Dresden, Germany. [54]Institute for Biology, Humboldt University of Berlin, Berlin, Germany. [55]Biomax Informatics AG, Planegg, Germany. [56]Harvard Medical School, Laboratory of Systems Pharmacology, Boston, MA, USA. [57]Department of Biology, University of Rome Tor Vergata, Rome, Italy. [58]Elsevier, Philadelphia, PA, USA. [59]Research Collaborations Unit, Elsevier, Jericho, VT, USA. [60]Institute for Computational Biomedicine, Heidelberg University, Heidelberg, Germany. [61]cBio Center, Divisions of Biostatistics and Computational Biology, Department of Data Sciences, Dana-Farber Cancer Institute, Boston, MA, USA. [62]Department of Cell Biology, Harvard Medical School, Boston, MA, USA. [63]Computer Science Department, University of Massachusetts Boston, Boston, MA, USA. [64]Clinical Bioinformatics Area, Fundación Progreso y Salud (FPS), Hospital Virgen del Rocio, Sevilla, Spain. [65]Computational Systems Medicine Group, Institute of Biomedicine of Seville (IBIS), Hospital Virgen del Rocio, Sevilla, Spain. [66]Bioinformatics in Rare Diseases (BiER), Centro de Investigación Biomédica en Red de Enfermedades Raras (CIBERER), FPS, Hospital Virgen del Rocío, Sevilla, Spain. [67]Department of Biochemistry, University of Nebraska-Lincoln, Lincoln, NE, USA. [68]Quadram Institute Bioscience, Norwich, UK. [69]Earlham Institute, Norwich, UK. [70]European Institute for Systems Biology and Medicine (EISBM), Vourles, France. [71]Cancer Research UK Cambridge Institute, University of Cambridge, Cambridge, UK. [72]Institute of Experimental Medicine and Systems Biology, Faculty of Medicine, RWTH, Aachen University, Aachen, Germany. [73]The Roslin Institute, University of Edinburgh, Edinburgh, UK. [74]University of Lausanne, Lausanne, Switzerland. [75]Helmholtz Zentrum München – German Research Center for Environmental Health, Institute of Computational Biology, Neuherberg, Germany. [76]Interdisciplinary Research Unit Mathematics and Life Sciences, University of Bonn, Bonn, Germany. [77]St. John's University College of Pharmacy and Health Sciences, Queens, NY, USA. [78]Department of Molecular Genetics, University of Toronto, Toronto, ON, Canada. [79]FPS/ELIXIR-es, Hospital Virgen del Rocío, Sevilla, Spain. [80]Institució Catalana de Recerca i Estudis Avançats (ICREA), Barcelona, Spain. [81]Systems Biology Institute, Tokyo, Japan. [82]Okinawa Institute of Science and Technology Graduate School, Okinawa, Japan.

