## [Review Process File · Molecular Systems Biology]

COVID-19 Disease Map, a computational knowledge repository of virus-host interaction mechanisms

Marek Ostaszewski, Anna Niarakis, Alexander Mazein, Inna Kuperstein, Robert Phair, Aurelio Ortega-Resendiz, Vidisha Singh, Sara Aghamiri, Marcio Acencio, Enrico Glaab, Andreas Ruepp, Gisela Fobo, Corinna Montrone, Barbara Brauner, Goar Frishman, Luis-Cristobal Monraz Gomez, Julia Somers, Matti Hoch, Shailendra K Gupta, Julia Scheel, Hanna Borlinghaus, Tobias Czauderna, Falk Schreiber, Arnau Montagud, Miguel Ponce de León, Akira Funahashi, Yusuke Hiki, Noriko Hiroi, Takahiro Yamada, Andreas Dräger, Alina Renz, Muhammad Naveez, Zsolt Bocksei, Francesco Messina, Daniela Börnigen, Liam Fergusson, Marta Conti, Marius Rameil, Vanessa Nakonecniij, Jakob Vanhoefer, Leonard Schmiester, Muying Wang, Emily Ackerman, Jason Shoemaker, Jeremy Zucker, Kristie Oxford, Jeremy Teuton, Ebru Kocakaya, Gokce Summak, Kristina Hanspers, Martina Kutmon, Susan Coort, Lars Eijssen, Friederike Ehrhart, D. A. B. Rex, Denise Slenter, Marvin Martens, Nhung Pham, Robin Haw, Bijay Jassal, Lisa Matthews, Marija Orlic-Milacic, Andrea Senff-Ribeiro, Karen Rothfels, Veronica Shamovsky, Ralf Stephan, Cristoffer Sevilla, Thawfeek Varusai, Jean-Marie Ravel, Rupsha Fraser, Vera Ortseifen, Silvia Marchesi, Piotr Gawron, Ewa Smula, Laurent Heirendt, Venkata Satagopam, Guanming Wu, Anders Riutta, Martin Golebiewski, Stuart Owen, Carol Goble, Xiaoming Hu, Rupert Overall, Dieter Maier, Angela Bauch, John Bachman, Benjamin Gyori, Carlos Vega, Valentin Groues, Miguel Vazquez, Pablo Porras, Luana Licata, Marta Iannuccelli, Francesca Sacco, Anastasia Nesterova, Anton Yuryev, Anita de Waard, Denes Turei, Augustin Luna, Ozgun Babur, Sylvain Soliman, Alberto Valdeolivas, Marina Esteban, Maria Peña-Chilet, Kinza Rian, Tomas Helikar, Bhanwar Lal Puniya, Dezso Modos, Agatha Treveil, Marton Olbei, Bertrand De Meulder, Stephane Ballereau, Aurelien Dugourd, Aurélien Naldi, Vincent Noel, Laurence Calzone, Chris Sander, Emek Demir, Tamas Korcsmaros, Tom Freeman, Franck Augé, Jacques Beckmann, Jan Hasenauer, Olaf Wolkenhauer, Egon Wilighagen, Alexander Pico, Chris Evelo, Marc Gillespie, Lincoln Stein, Henning Hermjakob, Peter D'Eustachio, Julio Saez-Rodriguez, Joaquin Dopazo, Alfonso Valencia, Hiroaki Kitano, Emmanuel Barillot, Charles Auffray, Rudi Balling, and Reinhard Schneider, and the COVID-19 Disease Map Community

DOI: [10.15252/msb.202110387](https://doi.org/10.15252/msb.202110387)

Corresponding author: Marek Ostaszewski (marek.ostaszewski@uni.lu)

Review Timeline:

Submission Date:	15th Apr 21
Editorial Decision:	11th May 21
Revision Received:	9th Jul 21
Editorial Decision:	12th Aug 21
Revision Received:	25th Aug 21
Accepted:	26th Aug 21

Editor: Jingyi Hou

Transaction Report:

Dear Dr Ostaszewski,

Thank you for submitting your work to Molecular Systems Biology. We have now heard back from

the three reviewers who agreed to evaluate your manuscript. As you will see below, the reviewers think the study is potentially interesting. They raise, however, a series of concerns, which we would ask you to address in a major revision.

The reviewers' recommendations are rather clear, and there is no need to reiterate their comments. All issues need to be satisfactorily addressed. In particular, the reviewers made constructive suggestions to improve the data presentation and clarity. In light of the concerns of Reviewer #3, we would ask you to edit the manuscript to make sure that the main results are sufficiently clear and easily accessible to the general audience of Molecular Systems Biology. As you may already know, our editorial policy allows in principle a single round of major revision, so it is essential to respond to the reviewers' comments that are as complete as possible. Please feel free to contact me in case you would like to discuss in further detail any of the issues raised by the reviewers.

On a more editorial level, please do the following:

- Please provide a .docx formatted version of the manuscript text (including legends for main figures, EV figures and tables). Please make sure that the changes are highlighted to be clearly visible.
- Please provide individual production quality figure files as .eps, .tif, .jpg (one file per figure).
- Please provide a .docx formatted letter INCLUDING the reviewers' reports and your detailed point-by-point responses to their comments. As part of the EMBO Press transparent editorial process, the point-by-point response is part of the Review Process File (RPF), which will be published alongside your paper.
- Please note that all corresponding authors are required to supply an ORCID ID for their name upon submission of a revised manuscript.
- We replaced Supplementary Information with Expanded View (EV) Figures and Tables that are collapsible/expandable online (see examples in <http://msb.embopress.org/content/11/6/812>). A maximum of 5 EV Figures can be typeset. EV Figures should be cited as 'Figure EV1, Figure EV2' etc... in the text and their respective legends should be included in the main text after the legends of regular figures.

Additional Tables/Datasets should be labeled and referred to as Table EV1, Dataset EV1, etc. Legends have to be provided in a separate tab in case of .xls files. Alternatively, the legend can be supplied as a separate text file (README) and zipped together with the Table/Dataset file.

For the figures and tables that you do NOT wish to display as Expanded View figures, they should be bundled together with their legends in a single PDF file called *Appendix*, which should start with a short Table of Content. Each legend should be below the corresponding Figure/Table in the Appendix. Appendix figures and tables should be referred to in the main text as: "Appendix Figure S1, Appendix Figure S2, Appendix Table S1" etc. See detailed instructions regarding expanded view here: <https://www.embopress.org/page/journal/17444292/authorguide#expandedview>.

-Before submitting your revision, primary datasets (and computer code, where appropriate) produced in this study need to be deposited in an appropriate public database (see <https://www.embopress.org/page/journal/17444292/authorguide#dataavailability>).

The accession numbers and database should be listed in a formal "Data Availability" section (placed after Materials & Method) that follows the model below (see also <https://www.embopress.org/page/journal/17444292/authorguide#dataavailability>). Please note that the Data Availability Section is restricted to new primary data that are part of this study.

Data availability

- We would encourage you to include the source data for figure panels that show essential quantitative information. Additional information on source data and instruction on how to label the files are available at < <https://www.embopress.org/page/journal/17444292/authorguide#sourcedata> >.

- All Materials and Methods need to be described in the main text. Please use 'Structured Methods', our new Materials and Methods format. According to this format, the Material and Methods section should include a Reagents and Tools Table (listing key reagents, experimental models, software and relevant equipment and including their sources and relevant identifiers) followed by a Methods and Protocols section in which we encourage the authors to describe their methods using a step-by-step protocol format with bullet points, to facilitate the adoption of the methodologies across labs. More information on how to adhere to this format as well as downloadable templates (.doc or .xls) for the Reagents and Tools Table can be found in our author guidelines: < <https://www.embopress.org/page/journal/17444292/authorguide#researcharticleguide> >. An example of a Method paper with Structured Methods can be found here: .

- Regarding data quantification:

Please ensure to specify the name of the statistical test used to generate error bars and P values, the number (n) of independent experiments (please specify technical or biological replicates) underlying each data point and the test used to calculate p-values in each figure legend. Discussion of statistical methodology can be reported in the materials and methods section, but figure legends should contain a basic description of n, P and the test applied. Graphs must include a description of the bars and the error bars (s.d., s.e.m.). Please also include scale bars in all microscopy images.

- Please provide a "standfirst text" summarizing the study in one or two sentences (approximately 250 characters, including space), three to four "bullet points" highlighting the main findings and a "synopsis image" (550px width and max 400px height, jpeg format) to highlight the paper on our homepage.

Here are a couple of examples:

<https://www.embopress.org/doi/10.15252/msb.20199356>

<https://www.embopress.org/doi/10.15252/msb.20209475>

<https://www.embopress.org/doi/10.15252/msb.209495>

When you resubmit your manuscript, please download our CHECKLIST

(<http://bit.ly/EMBOPressAuthorChecklist>) and include the completed form in your submission.

Please note that the Author Checklist will be published alongside the paper as part of the transparent process (<https://www.embopress.org/page/journal/17444292/authorguide#transparentprocess>).

If you feel you can satisfactorily deal with these points and those listed by the referees, you may wish to submit a revised version of your manuscript. Please attach a covering letter giving details of the way in which you have handled each of the points raised by the referees. A revised manuscript will be once again subject to review and you probably understand that we can give you no guarantee at this stage that the eventual outcome will be favorable.

Kind regards,
Jingyi

Jingyi Hou
Editor
Molecular Systems Biology

Reviewer #1:

The authors describe the "COVID-19 Disease Map", resulting from a massive community collaboration. The map is open access and provides a graphical and interactive view of the molecular mechanisms resulting from SARS-CoV-2 infection. It provides curated computational diagrams and models of molecular mechanisms involved in the disease, and incorporates a large amount of different data types, including pathway collections, literature information (using NLP) and interaction databases. The map is both human and machine readable and is a valuable resource for the scientific community in helping us better understand COVID-19.

The manuscript covers both the map and its contents, as well as the community and effort that went into realizing the map. The focus is mainly on these aspects, rather than on major new findings extracted from the map. I think this is fine, as the map itself is the key resource, and others can make use of it for further in-depth explorations. In addition, the manuscript does outline a few examples of how the map can be informative about new processes related to SARS-CoV-2 infection.

Overall, the manuscript is well-written, but some figures (and related text) are confusing or have errors.

I support the publication of this important resource but ask that the points below are addressed first:

- The COVID-19 Disease Map is referred to as "the Map" throughout the manuscript. While this is self-explanatory, it would be nice to define this shorthand early on. E.g. "The COVID-19 Disease Map (The Map)".

- First sentence of intro: "already resulted in" -> "has already resulted in"

- Section 2. Regarding cell-specific versions of the map: Is this not possible at all yet? Or could the authors start out with one/two cell types and prepare for this expansion into more cell types?

- Section 2.1 describes various biological processes relevant to SARS-CoV-2 but it's hard to contextualize these observations when the authors make specific references to the events described in the map without providing an image of the relevant map. Most likely, the reader is meant to use Table S1 as a guide for this purpose, which is fine, but the section would benefit from clearly noting this.

- Section 3.1, first paragraph.

The following needs rewording: "Projection of data on the Map may provide their better understanding...".

Further, this sentence could be rewritten to flow better: "Visualisation of omics datasets on the Map diagrams creates overlays allowing to interpret specific conditions, like disease severity or cell types."

- Discussion: "It offers a shared mental map..." What is a mental map?

- Discussion: "serve as a blueprint for a formalised and standardised streamline of well-defined tasks."

To my knowledge "streamline" cannot be used as a noun (other than in physics for a very specific purpose).

- Figure 1:

"Antibodies production" -> Antibody production

"Antigen-Presenting Cell" -> Capitalization does not match rest of figure.

T-cell or T cell? Pick one for consistency.

- Figure 3: The legend states that "The distribution of the elements is for illustrative reference and does not necessarily indicate either a unique/static interplay of these elements or an unvarying progression." but this results in a very confusing figure. The way different categories are stacked and every category's background color going from a light to dark gradient visually implies a connection between these groupings. Additionally, some parts appear to fit together from a progression perspective, while others do not. The top row appears to focus on severity and the row below on infection progression, which may or may not be related... There could be more clear boundaries to reflect which are independent categories (or if all are).

Also, it is unclear what the "recovery/death" axis is meant to represent, especially because it's not connected to any other information.

If this cannot be made clearer, a table may be more appropriate, as the current layout could lead to misinformation. For example, the current layout appears to suggest that an asymptomatic infection does not involve an immune response.

Why is there a label that says Host in bold, followed by "coagulation" and "Cytokine release..."

Prophilaxis -> Prophylaxis

- In Figure 4 and the related supplementary figures, it would be helpful to include legends within the figures that explain the color code. Related to this point, Figs 4C and S5 have some blue nodes that are connected to just one other node (such as IL2, SRC, G3BP1, BIRC2, etc.) even though the

legend text states that blue nodes are proteins with two neighbors. Was that meant to say "one or two neighbors" or should these nodes be updated to a different color?

- In the caption for Figure 4, spell out "neighbors" in all cases (implied currently).

- Edge directionality across Fig 4 and supplementary figures are inconsistent and there isn't a clear indication whether these differences are actually meant to reflect distinctions between these networks. For instance, Fig 4A is an undirected network, but pathway-protein edges in the rest of the networks are directed and their directionality changes between networks. Figs S1 and S2 have directed edges that go from protein nodes to pathway nodes but Figs 4B, 4C, S3, S4, and S5 have directed edges in the opposite direction (from pathway nodes to protein nodes). Is there a meaningful distinction between these choices of edge types that are not explained in the legend?

- Supp Text 3 describes how interactions not found in the current diagrams are used to identify new crosstalk and upstream regulators and Figs 4B, 4C, S3, S4, and S5 are presented as examples. However, in these network views, there is no visual way to differentiate these external proteins and interactions from the ones available in the disease map. This makes it hard to pinpoint the interactions and proteins that are driving these new crosstalks. Using a different color/node shape/edge type for these nodes and edges would make interpretation of the results easier.

- "Novel regulators of key pathway proteins" section focuses on NFE2L2-HMOX1 axis and describes NFE2L2 as a novel regulator. But the network in Fig 4C is dominated by many other pathways and proteins that do not seem to be directly related to this axis. "Coagulopathy pathway" and 4 proteins connected to it aren't even a part of the connected component containing HMOX1. How were these other proteins and pathways selected to be included in this view? Was it because they represent other "novel regulators" that are not mentioned in the text?

- It appears the only difference between Fig 4B and Fig S4 is an "Other diagrams" node that connects AKT1 and TNF. Similarly, it appears the main distinction between Fig 4C and Fig S5 is an "Other diagrams" node that connects TRAF6 to RHEB. The "Other diagrams" nodes provide such a vague description that I am not sure it significantly differentiates these network views from the ones in the main figure. Could these "other diagrams" be expanded to describe the actual processes that drive these connections?

- I suspect the sentence "As highlighted in Figure 4, our manually curated pathway included some of the most active TFs after SARS-CoV-2 infection, such as STAT1, STAT2, IRF9 and NFKB1." in section 3.3 is actually referring to Fig 5.

Similarly, the sentence "Results of the Apoptosis pathway analysis can be seen in Figure 5 and Supplementary Table S2." in section 3.4 is referring to Fig 6.

- The legend of Fig 6 states that metabolites are represented with circles, however as far as I can see, viral proteins are the only nodes represented with circles in this network. Are viral proteins somehow treated as metabolites in this representation or should the legend be updated? Additionally, the Orf6 node has an orange ring around it that is not seen in any other node or explained in the legend. Is there a significance to it?

- The legend of Fig 6 is very poor resolution

- Fig 6 caption - "normal lung cells". What does this mean? Uninfected? What type of cell line?

- Fig 5 is not legible due to very poor resolution, so I cannot comment on this.

Reviewer #2:

In this paper, the authors present the COVID-19 Disease Map a powerful resource to gain insight into the molecular mechanisms of COVID-19 and SARS-CoV-2 infection.

I found interesting the graphical representation of the COVID-19 Disease Map as well as the integration with most of the common XML-based formats making easier the graph-based and disease modelling analysis.

I have only some minor comments that may strengthen both manuscript and the COVID-19 Disease Map platform.

Manuscript:

-The resolution of figure 5 should be improved.

-figure 3 is duplicated (seems to also be present in figure 2)

COVID-19 Disease Map platform:

-Despite it not being reported in the manuscript, the platform permits to load the GSEA plugin. However, it doesn't seem to work properly when the results are downloaded (pvalues and other information are missing). It should be corrected or removed.

-Lighter colour should be used when the COVID19 scRNAseq overlay is loaded, the actual colours are too dark making it impossible to read the underlying names (in the submap included in the manuscript).

-It would be useful to implement the autocomplete function also for the other tabs (drug, chemical and miRNA)

Reviewer #3:

The manuscript submitted by Ostaszewski et al, entitled "COVID-19 Disease Map, a computational knowledge repository of virus-host interaction mechanisms" describes an impressive collaborative effort across many labs to curate biological pathway models that are relevant for the SARS-CoV-2 viral replication cycle and affected host processes. The authors employed data mining as well as manual curation to build these pathway models and build an infrastructure that eases integration of these pathway models into omic data analysis workflows that relate to SARS-CoV-2 infections. Of note, the outcome of this effort is not a new database or webserver on collected pathways and mechanisms around SARS-CoV-2 infections but rather the curated models, which can be downloaded, explored, and visualized at various existing websites such as Reactome. Given the explosion of publications on biological findings that relate to SARS-CoV-2 infections to fight the current global pandemic, efforts aimed at collecting and integrating information on the published data and mechanisms that underlie SARS-CoV-2 infections is extremely helpful in ensuring that no or only few published findings are lost and new testable hypotheses can be formed based on the integration of these. I am not aware of another similar effort. Developed strategies and routines for this collaborative effort are likely applicable to other biological emergencies that might arise in the future.

I think the results from this study will be of interest to experimentalists who seek to obtain a comprehensive overview of published findings in the field of SARS-CoV-2 biology, who seek to

integrate their data with pathway models for interpretation, i.e. of identified differentially expressed genes, and for computational biologists who aim to integrate SARS-CoV-2 related data for candidate selection and hypothesis generation.

Major points:

The results part is in many parts written in very technical and abstract terms such that it is probably only understandable by experts in the field of pathway modeling. This is especially true for section 1 of the results, i.e. already the title of section 1 is very abstract and technical. Technical terms as well as tools, webservices, etc that were used are not defined or explained. Maybe some of the technicalities can be put into a Methods section and some sentences can be added or reformulated to describe in more lenient terms, understandable by biologists, what the approach taken to curate these pathway models was about. I wonder whether some of the approaches taken for data curation could be better visualized, i.e. with screenshots of the tools used. The two case studies presented are very interesting and helpful to see the potential of the infrastructure that has been built to integrate the pathway models with experimental data. However, looking up the corresponding code and documentation provided on the gitlab website, I was not able to find code that relates to the case study 3.3 and I wasn't able to understand whether the code named Hipathia actually reproduced the analyses provided in case study 3.4. It would be really helpful, if the authors could make more effort in providing better documented code recipes to start with for certain analysis tasks like the ones illustrated in 3.3 and 3.4.

Minor points:

Introduction: I think that an inappropriate long part of the introduction, i.e. about half of it, is dedicated to summarizing the approach and results of this study. I would advice to shorten this to a smaller paragraph. Furthermore, the authors could expand on the need to integrate published findings on SARS-CoV-2 infections by, i.e. showing the increase in publications on this matter (this is to some extent mentioned in the results section but might be better suited for the introduction) and reviewing possible other existing resources aimed at collecting SARS-CoV-2 findings, i.e. IntAct or PDB, and where they fall short. And which information on SARS-CoV-2 relevant pathways existed in Reactome and Wikipathways before this collaborative effort was initiated?

Case study 3.3. For the better understanding by the readership, it would be helpful if the authors could introduce the case study with the biological question that they were trying to answer in this analysis, explain what the different tools do that are applied here, i.e. VIPER and DoRothEA, and be more precise about the data that is being used. I.e. I assume you used data of SARS-CoV-2 infected and non-infected Calu-3 cell lines but you do not specify the latter. Also, in this paragraph you refer to Figure 4 but I assume it should be Figure 5? Figure 5 was not readable because the resolution was too low, unfortunately.

Case study 3.4. The authors refer to an overactivation of several circuits. Can you please clarify why you interpret the upregulation as an overactivation and overexpression? To the best of my understanding, two conditions, normal and infected cells were analyzed with respect to whether genes involved in apoptosis are differentially expressed or upregulated in the infected versus normal cells. Unless I miss something, this is not enough to interpret the upregulation as an overactivation. This would require comparison to expression levels under different conditions when apoptosis is being triggered. Also, could it be that the reference to Figure 5 should have rather been Figure 6?

Data availability section: The WikiPathways link does not seem to point to the right location, please check. The Reactome link does not work.

Discussion: The value of the study could be more apparent, if the authors were able to more specifically discuss which novelties among the approach taken here can be used for future such efforts and which other resources apart from the pathway models are novel. It remains unclear whether any of the infrastructure build to integrate the pathway models with other data is novel or based on combining existing tools.

Point-by-point response to Reviewers

Major revision of the manuscript “COVID-19 Disease Map, a computational knowledge repository of virus-host interaction mechanisms” by Ostaszewski et al.

We would like to thank the Reviewers for their thorough evaluation of our manuscript and their detailed and extensive comments. Addressing their remarks significantly improved the manuscript, and we hope we were able to answer all their questions and suggestions sufficiently. Below is our point-by-point response to the raised issues.

On behalf of the authors,
Marek Ostaszewski, Anna Niarakis, Alexander Mazein and Inna Kuperstein

Reviewer #1:

The authors describe the "COVID-19 Disease Map", resulting from a massive community collaboration. The map is open access and provides a graphical and interactive view of the molecular mechanisms resulting from SARS-CoV-2 infection. It provides curated computational diagrams and models of molecular mechanisms involved in the disease, and incorporates a large amount of different data types, including pathway collections, literature information (using NLP) and interaction databases. The map is both human and machine readable and is a valuable resource for the scientific community in helping us better understand COVID-19.

The manuscript covers both the map and its contents, as well as the community and effort that went into realizing the map. The focus is mainly on these aspects, rather than on major new findings extracted from the map. I think this is fine, as the map itself is the key resource, and others can make use of it for further in-depth explorations. In addition, the manuscript does outline a few examples of how the map can be informative about new processes related to SARS-CoV-2 infection.

Overall, the manuscript is well-written, but some figures (and related text) are confusing or have errors.

I support the publication of this important resource but ask that the points below are addressed first:

Remark:

- The COVID-19 Disease Map is referred to as "the Map" throughout the manuscript. While this is self-explanatory, it would be nice to define this shorthand early on. E.g. "The COVID-19 Disease Map (The Map)".

Response: We have added this definition at the beginning of the article (in the Abstract, 3rd sentence, and in the Introduction, 2nd paragraph, 1st sentence). To be more specific, we replaced “the Map” with “the C19DMap”.

Remark:

- First sentence of intro: "already resulted in" -> "has already resulted in"

Response: We have made the requested change.

Remark:

- Section 2. Regarding cell-specific versions of the map: Is this not possible at all yet? Or could the authors start out with one/two cell types and prepare for this expansion into more cell types?

Response:

Cell specificity is indeed an important aspect of the project. To the best of our knowledge so far, researchers have experimentally identified close to 20 different cell types as susceptible to SARS-CoV-2 infection. To make the curation effort scalable, at this stage, we propose to project expression data obtained from cell-specific experiments onto C19DMap to infer affected pathways, as for instance routes of SARS-CoV-2 entry, like endocytosis or direct membrane fusion. In the manuscript, we introduce a new subsection in the Results part (3.3 Case study: Analysis of cell-specific mechanisms using single cell expression data, page 14) to illustrate this approach.

Remark:

- Section 2.1 describes various biological processes relevant to SARS-CoV-2 but it's hard to contextualize these observations when the authors make specific references to the events described in the map without providing an image of the relevant map. Most likely, the reader is meant to use Table S1 as a guide for this purpose, which is fine, but the section would benefit from clearly noting this.

Response: Table S1 is now included in the manuscript as Expanded View Table (Table EV1). Respective paragraphs of the overview now mention diagrams from Table EV1, making a direct reference to the contents of C19DMap.

Remark:

- Section 3.1, first paragraph.

The following needs rewording: "Projection of data on the Map may provide their better understanding...".

Further, this sentence could be rewritten to flow better: "Visualisation of omics datasets on the Map diagrams creates overlays allowing to interpret specific conditions, like disease severity or cell types."

Response: We have corrected this part of the text following the Reviewer's suggestions, first paragraph in Section 3.1 on page 13.

Remark:

- Discussion: "It offers a shared mental map..." What is a mental map?

Response: For clarity, the sentence "It offers a shared mental map for understanding the dynamic nature of the disease at the molecular level and its propagation at a systemic level." has been changed to "It offers insights into the dynamic nature of the disease at the molecular level and its propagation at a systemic level. " on page 18, last paragraph, 2nd sentence.

Remark:

- Discussion: "serve as a blueprint for a formalised and standardised streamline of well-defined tasks."

To my knowledge "streamline" cannot be used as a noun (other than in physics for a very specific purpose).

Response: We have corrected this part of the text following the Reviewer's suggestions, with the word "streamline" replaced by "workflow".

Remark:

- Figure 1:

"Antibodies production" -> Antibody production

"Antigen-Presenting Cell" -> Capitalization does not match the rest of the figure.

T-cell or T cell? Pick one for consistency.

Response: It seems the Reviewer was referring to Figure 2, we have now corrected the figure legend following the Reviewer's suggestions.

Remark:

- Figure 3: The legend states that "The distribution of the elements is for illustrative reference and does not necessarily indicate either a unique/static interplay of these elements or an unvarying progression." but this results in a very confusing figure. The way different categories are stacked and every category's background color going from a light to dark gradient visually implies a connection between these groupings. Additionally, some parts appear to fit together from a progression perspective, while others do not. The top row appears to focus on severity and the row below on

infection progression, which may or may not be related... There could be more clear boundaries to reflect which are independent categories (or if all are).

Also, it is unclear what the "recovery/death" axis is meant to represent, especially because it's not connected to any other information.

If this cannot be made clearer, a table may be more appropriate, as the current layout could lead to misinformation. For example, the current layout appears to suggest that an asymptomatic infection does not involve an immune response.

Why is there a label that says Host in bold, followed by "coagulation" and "Cytokine release..."

Prophilaxis -> Prophylaxis

Response: Figure 3 and its description have been improved to address the issues raised by the Reviewer.

Remark:

- In Figure 4 and the related supplementary figures, it would be helpful to include legends within the figures that explain the color code. Related to this point, Figs 4C and S5 have some blue nodes that are connected to just one other node (such as IL2, SRC, G3BP1, BIRC2, etc.) even though the legend text states that blue nodes are proteins with two neighbors. Was that meant to say "one or two neighbors" or should these nodes be updated to a different color?

Response: The figures are now Expanded View Figures. We have introduced the requested legends. Indeed, the colour code meant "one or two neighbors", and the legends have been updated to mention this.

Remark:

- In the caption for Figure 4, spell out "neighbours" in all cases (implied currently).

Response: We have made the requested change.

Remark:

- Edge directionality across Fig 4 and supplementary figures are inconsistent and there isn't a clear indication whether these differences are actually meant to reflect distinctions between these networks. For instance, Fig 4A is an undirected network, but pathway-protein edges in the rest of the networks are directed and their directionality changes between networks. Figs S1 and S2 have directed edges that go from protein nodes to pathway nodes but Figs 4B, 4C, S3, S4, and S5 have directed edges in the opposite direction (from pathway nodes to protein nodes). Is there a meaningful distinction between these choices of edge types that are not explained in the legend?

- Supp Text 3 describes how interactions not found in the current diagrams are used to identify new crosstalk and upstream regulators and Figs 4B, 4C, S3, S4, and S5 are presented as examples. However, in these network views, there is no visual way to differentiate these external proteins and interactions from the ones available in the disease map. This makes it hard to pinpoint the interactions and proteins that are driving these new crosstalks. Using a different color/node shape/edge type for these nodes and edges would make interpretation of the results easier.

Response: Indeed, the directionality of the edges was illustrated inaccurately. Figure 4 and Figures EV1-5 have now been updated in the following manner:

- Edges between molecules and pathways are undirected
- New crosstalks (edges between molecules) are directed
- New regulators are marked with solid black border

All these details are provided in the figure legends.

Remark:

- "Novel regulators of key pathway proteins" section focuses on NFE2L2-HMOX1 axis and describes NFE2L2 as a novel regulator. But the network in Fig 4C is dominated by many other pathways and proteins that do not seem to be directly related to this axis. "Coagulopathy pathway" and 4 proteins connected to it aren't even a part of the connected component containing HMOX1. How were these other proteins and pathways selected to be included in this view? Was it because they represent other "novel regulators" that are not mentioned in the text?

Response: Indeed, Fig 4C represents all novel regulators and their targets identified by the approach. We chose some examples as discussing the entire diagram is out of scope of this article. The text of the article now emphasises this, and indicates how to access the full contents of the crosstalk diagrams. "Coagulopathy pathway" subnetwork is isolated because even though it has no crosstalk interactions (for this reason it is excluded from Fig 4A and 4B), it is a target of novel regulators.

Remark:

- It appears the only difference between Fig 4B and Fig S4 is an "Other diagrams" node that connects AKT1 and TNF. Similarly, it appears the main distinction between Fig 4C and Fig S5 is an "Other diagrams" node that connects TRAF6 to RHEB. The "Other diagrams" nodes provide such a vague description that I am not sure it significantly differentiates these network views from the ones in the main figure. Could these "other diagrams" be expanded to describe the actual processes that drive these connections?

Response: Indeed, the description "Other diagrams" is vague and misleading. This group represented two WikiPathways diagrams, WP4891 and WP5017, which are not included in Figure 4 and Table EV1 because they carry insufficient mechanistic detail in comparison to the core diagrams of C19DMap. For the sake of clarity, we excluded these diagrams from our analysis. Consequently,

Figures EV4 and EV5 were removed, as the node "Other diagrams" was their only difference from Fig 4B and 4C, respectively.

Remark:

- I suspect the sentence "As highlighted in Figure 4, our manually curated pathway included some of the most active TFs after SARS-CoV-2 infection, such as STAT1, STAT2, IRF9 and NFKB1." in section 3.3 is actually referring to Fig 5.

Similarly, the sentence "Results of the Apoptosis pathway analysis can be seen in Figure 5 and Supplementary Table S2." in section 3.4 is referring to Fig 6.

Response: We have corrected these inaccurate figure references.

Remark:

- The legend of Fig 6 states that metabolites are represented with circles, however as far as I can see, viral proteins are the only nodes represented with circles in this network. Are viral proteins somehow treated as metabolites in this representation or should the legend be updated? Additionally, the Orf6 node has an orange ring around it that is not seen in any other node or explained in the legend. Is there a significance to it?

- The legend of Fig 6 is very poor resolution

- Fig 6 caption - "normal lung cells". What does this mean? Uninfected? What type of cell line?

- Fig 5 is not legible due to very poor resolution, so I cannot comment on this.

Response: The Figures and their legends have been improved following the suggestions of the Reviewer. As in the process of revision new versions of the diagrams and new datasets became available, these Figures are also improved to better illustrate corresponding use-cases.

Reviewer #2:

In this paper, the authors present the COVID-19 Disease Map a powerful resource to gain insight into the molecular mechanisms of COVID-19 and SARS-CoV-2 infection.

I found interesting the graphical representation of the COVID-19 Disease Map as well as the integration with most of the common XML-based formats making easier the graph-based and disease modelling analysis.

I have only some minor comments that may strengthen both manuscript and the COVID-19 Disease Map platform.

Manuscript:

Remark:

The resolution of figure 5 should be improved.

Response: We have made the requested change.

Remark:

figure 3 is duplicated (seems to also be present in figure 2)

Response: As a mistake in the submission process, Figure 3 was duplicated. Moreover, we improved Figures 2 and 3 to distinctively represent two facets of the COVID-19 pathology - its molecular composition and its course.

COVID-19 Disease Map platform:

Remark:

Despite it not being reported in the manuscript, the platform permits to load the GSEA plugin.

However, it doesn't seem to work properly when the results are downloaded (pvalues and other information are missing). It should be corrected or removed.

Response: The description of the GSEA plugin function is now mentioned in the manuscript. The functionality of the plugin is described in detail in the user guide of the MINERVA Platform (<https://covid.pages.uni.lu/minerva-guide/>). The functionality of the plugin has been corrected, p-values are now exported correctly.

Remark:

Lighter colour should be used when the COVID19 scRNAseq overlay is loaded, the actual colours are too dark making it impossible to read the underlying names (in the submap included in the manuscript).

Response: Colour profiles of the available overlays have been improved.

Remark:

It would be useful to implement the autocomplete function also for the other tabs (drug, chemical and miRNA)

Response: This is an error of the autocomplete function that is already implemented in the MINERVA Platform. We have raised this issue using the project tracker (<https://git-r3lab.uni.lu/minerva/core/-/issues/1520>).

Reviewer #3:

The manuscript submitted by Ostaszewski et al, entitled "COVID-19 Disease Map, a computational knowledge repository of virus-host interaction mechanisms" describes an impressive collaborative effort across many labs to curate biological pathway models that are relevant for the SARS-CoV-2 viral replication cycle and affected host processes. The authors employed data mining as well as manual curation to build these pathway models and build an infrastructure that eases integration of these pathway models into omic data analysis workflows that relate to SARS-CoV-2 infections. Of note, the outcome of this effort is not a new database or webserver on collected pathways and mechanisms around SARS-CoV-2 infections but rather the curated models, which can be downloaded, explored, and visualized at various existing websites such as Reactome. Given the explosion of publications on biological findings that relate to SARS-CoV-2 infections to fight the current global pandemic, efforts aimed at collecting and integrating information on the published data and mechanisms that underlie SARS-CoV-2 infections is extremely helpful in ensuring that no or only few published findings are lost and new testable hypotheses can be formed based on the integration of these. I am not aware of another similar effort. Developed strategies and routines for this collaborative effort are likely applicable to other biological emergencies that might arise in the future.

I think the results from this study will be of interest to experimentalists who seek to obtain a comprehensive overview of published findings in the field of SARS-CoV-2 biology, who seek to integrate their data with pathway models for interpretation, i.e. of identified differentially expressed genes, and for computational biologists who aim to integrate SARS-CoV-2 related data for candidate selection and hypothesis generation.

Major points:

Remark:

The results part is in many parts written in very technical and abstract terms such that it is probably only understandable by experts in the field of pathway modeling. This is especially true for section 1 of the results, i.e. already the title of section 1 is very abstract and technical. Technical terms as well as tools, webservices, etc that were used are not defined or explained. Maybe some of the technicalities can be put into a Methods section and some sentences can be added or reformulated to describe in more lenient terms, understandable by biologists, what the approach taken to curate these pathway models was about. I wonder whether some of the approaches taken for data curation could be better visualized, i.e. with screenshots of the tools used.

Response: We have followed the suggestion of the Reviewer to improve the overall clarity and the narrative of the article. The text was revised and technical details were moved to the Materials and Methods section.

Remark:

The two case studies presented are very interesting and helpful to see the potential of the infrastructure that has been built to integrate the pathway models with experimental data. However, looking up the corresponding code and documentation provided on the gitlab website, I was not able to find code that relates to the case study 3.3 and I wasn't able to understand whether the code named Hipathia actually reproduced the analyses provided in case study 3.4. It would be really helpful, if the authors could make more effort in providing better documented code recipes to start with for certain analysis tasks like the ones illustrated in 3.3 and 3.4.

Response: In Materials and Methods we have provided pointers to code snippets as well as instructions on how to reproduce the results of all three use cases, including the newly added one (pages 24-26). For the case study in section 3.3 (new) the code allows to reproduce the expression profiles of single cells and to map them to C19DMap on the MINERVA Platform. For the case study in section 3.4 (former 3.3) the code allows to generate the input files for the webservice of HiPathia, and the instructions on how to run the analysis are provided. For the case study in section 3.5 (former 3.4) the link to the Jupyter notebooks is provided.

Minor points:

Remark:

Introduction: I think that an inappropriate long part of the introduction, i.e. about half of it, is dedicated to summarizing the approach and results of this study. I would advice to shorten this to a smaller paragraph. Furthermore, the authors could expand on the need to integrate published findings on SARS-CoV-2 infections by, i.e. showing the increase in publications on this matter (this is to some extent mentioned in the results section but might be better suited for the introduction) and reviewing possible other existing resources aimed at collecting SARS-CoV-2 findings, i.e. IntAct or PDB, and where they fall short. And which information on SARS-CoV-2 relevant pathways existed in Reactome and Wikipathways before this collaborative effort was initiated?

Response: We shortened the introduction and improved its narrative. We also added the information on the number of articles published in relation to COVID-19, as reported by PubMed (query: "covid-19[Title/Abstract] or sars-cov-2[Title/Abstract]"). The introduction now mentions PDB and IMEx interaction databases as high focus, limited scope resources, which indeed supports the argument of using the systems biology approach. Interestingly, the visualisation platform of the map enables display of viral proteins where their PDB structure is available. We have improved the manual to instruct users about this possibility. All content, including the Reactome and WikiPathways diagrams, was built *de novo*. This is now explicitly indicated in the article.

Remark:

Case study 3.3. For the better understanding by the readership, it would be helpful if the authors could introduce the case study with the biological question that they were trying to answer in this analysis, explain what the different tools that are applied here, i.e. VIPER and DoRothEA, and be more precise about the data that is being used. I.e. I assume you used data of SARS-CoV-2 infected and non-infected Calu-3 cell lines but you do not specify the latter. Also, in this paragraph you refer to Figure 4 but I assume it should be Figure 5? Figure 5 was not readable because the resolution was too low, unfortunately.

Response: The description of the case study in section 3.4 (former 3.3) was improved (page 15). The technical part was moved to Materials and Methods (page 23). Cell lines used in the analysis are now clearly specified (“SARS-CoV-2 infected Calu-3 human lung adenocarcinoma cell line and controls”). The resolution of Figure 5 was improved, the reference was corrected.

Remark:

Case study 3.4. The authors refer to an overactivation of several circuits. Can you please clarify why you interpret the upregulation as an overactivation and overexpression? To the best of my understanding, two conditions, normal and infected cells were analyzed with respect to whether genes involved in apoptosis are differentially expressed or upregulated in the infected versus normal cells. Unless I miss something, this is not enough to interpret the upregulation as an overactivation. This would require comparison to expression levels under different conditions when apoptosis is being triggered. Also, could it be that the reference to Figure 5 should have rather been Figure 6?

Response: The description of the case study in section 3.5 (former 3.4) was improved. The resolution of Figure 5 was improved, the reference was corrected. The HiPathia algorithm used to produce Figure 6 serves as a modelling pathway tool, it takes not only gene expression but also the interactions between the nodes, dividing the pathways into all possible receptor-effector circuits and measuring the activation of each circuit, as well as coloring the nodes with the differentially expressed genes belonging to each node. The comparison of activation levels in SARS-CoV-2 vs Controls is what enables us to observe an upregulation/downregulation of certain circuits, or maybe an overall activation/inhibition in case all the circuits appear as down/up-regulated versus the controls.

Remark:

Data availability section: The WikiPathways link does not seem to point to the right location, please check. The Reactome link does not work.

Response: The WikiPathways link was corrected. The link to the Reactome part of the COVID-19 Disease Map is correct, but the navigation through its landing page may not be intuitive. To address this remark, the user guide of the Reactome (<https://covid.pages.uni.lu/reactome-guide/>) was improved.

Remark:

Discussion: The value of the study could be more apparent, if the authors were able to more specifically discuss which novelties among the approach taken here can be used for future such efforts and which other resources apart from the pathway models are novel. It remains unclear whether any of the infrastructure built to integrate the pathway models with other data is novel or based on combining existing tools.

Response: We have revised the Discussion section by highlighting the added value of the project, focusing on community-level biocuration, establishing an interoperable and computational framework, and following the FAIR principles in our work.

Thank you for sending us your revised manuscript. We have now heard back from the three

reviewers who were asked to evaluate your study. As you will see the reviewers are overall satisfied with the modifications made and think that the study is now suitable for publication.

Before we can formally accept your manuscript, we would ask you to address the following issues:

1. Please address the remaining minor issues raised by Reviewer #1.

On a more editorial level:

2. Please provide up to five keywords and incorporate them in the main text.

3. 'Goar Frishman', 'Julia Somers', 'Friederike Ehrhart' are spelled incorrectly/differently in the Author contribution section and in the author list. Please double-check it and correct them.

4. Please rename 'Competing interest' to "Conflict of Interest".

5. Please upload Table EV1 and Table EV2 as separate files. Remove them from the manuscript file.

6. Please add contact information for the corresponding author(s) in the manuscript text.

7. Checklist: please fill out Box 18-20.

8. Our journal encourages inclusion of *data citations in the reference list* to directly cite datasets that were re-used and obtained from public databases. Data citations in the article text are distinct from normal bibliographical citations and should directly link to the database records from which the data can be accessed. In the main text, data citations are formatted as follows: "Data ref: Smith et al, 2001". In the Reference list, data citations must be labeled with "[DATASET]". A data reference must provide the database name, accession number/identifiers and a resolvable link to the landing page from which the data can be accessed at the end of the reference. Further instructions are available at .

9. I have slightly modified the synopsis text (see attached). Please let me know if it is fine as is or if you would like to introduce further modifications.

When you resubmit your manuscript, please download our CHECKLIST

(<https://bit.ly/EMBOPressAuthorChecklist> and include the completed form in your submission.

Please note that the Author Checklist will be published alongside the paper as part of the transparent process

(<https://www.embopress.org/page/journal/17444292/authorguide#transparentprocess>

Click on the link below to submit your revised paper.

Link Not Available

Thank you for submitting this paper to Molecular Systems Biology. I look forward to receiving your revised manuscript soon.

Kind regards,

Jingyi
Jingyi Hou
Editor
Molecular Systems Biology

Reviewer #1:

The new case study with single-cell expression data is an informative addition and I have one minor suggestion related to this: The section concludes with the observation that "Enrichment analysis of diagrams indicated that mitochondrial dysfunction, apoptosis, and inflammasome activation were dysregulated in infected enterocytes." The description of this analysis with one sentence feels incomplete, especially since there is no accompanying figure or table that provides quantitative context for these results and it only focuses on one of the cell types without an explanation on why this was chosen. However, based on the methods text, I am under the impression that this reflects the results obtained by the GSEA plugin available as part of the map. If that is the case, this could be more clearly expressed in the results section to help direct the readers find the relevant result list within the map portal.

The legend of Figure 6 is updated to say "Significantly deregulated circuits in infected cells are highlighted by red arrows." instead of "Significantly deregulated circuits are highlighted by color arrows (red: activated in infected cells)." Keeping this as "color arrows" (or "colored arrows") and specifying the "color-direction of change" mapping might have been better, especially since the updated version of the figure does not contain any red edges/arrows. Also, make sure that "deregulated" is really the correct choice of word here (as opposed to e.g. "regulated" or "differentially regulated")?

Reviewer #2:

The authors have adequately addressed many of my concerns, however, with Google Chrome (version 92 on Ubuntu) the red overlay still hides the text. It is noted that this effect is mitigated in Firefox.

Overall, I think that the revised version improved and it will be a widely used resource for the community.

The paper is now suitable for publication, in my opinion.

Reviewer #3:

The reviewers' concerns and remarks have been addressed to my full satisfaction. The manuscript can be published as is.

Point-by-point response to Reviewers comments on the major revision manuscript

Second revision of the manuscript "COVID-19 Disease Map, a computational knowledge repository of virus-host interaction mechanisms" by Ostaszewski et al.

We would like to thank the Reviewers for their comments on the major revision of our manuscript. We are happy that they found our revision to sufficiently address their remarks. Below is our point-by-point response to the remaining issues.

On behalf of the authors,
Marek Ostaszewski, Anna Niarakis, Alexander Mazein and Inna Kuperstein

Reviewer #1:

Remark:

The new case study with single-cell expression data is an informative addition and I have one minor suggestion related to this: The section concludes with the observation that "Enrichment analysis of diagrams indicated that mitochondrial dysfunction, apoptosis, and inflammasome activation were dysregulated in infected enterocytes." The description of this analysis with one sentence feels incomplete, especially since there is no accompanying figure or table that provides quantitative context for these results and it only focuses on one of the cell types without an explanation on why this was chosen. However, based on the methods text, I am under the impression that this reflects the results obtained by the GSEA plugin available as part of the map. If that is the case, this could be more clearly expressed in the results section to help direct the readers find the relevant result list within the map portal.

Response: We have added an explanation at the end of the paragraph following the Reviewer's suggestions.

Remark:

The legend of Figure 6 is updated to say "Significantly deregulated circuits in infected cells are highlighted by red arrows." instead of "Significantly deregulated circuits are highlighted by color arrows (red: activated in infected cells)." Keeping this as "color arrows" (or "colored arrows") and specifying the "color-direction of change" mapping might have been better, especially since the updated version of the figure does not contain any red edges/arrows. Also, make sure that "deregulated" is really the correct choice of word here (as opposed to e.g. "regulated" or "differentially regulated")?

Response: We have corrected the caption of Figure 6 following the Reviewer's suggestions.

Reviewer #2:

The authors have adequately addressed many of my concerns, however, with Google Chrome (version 92 on Ubuntu) the red overlay still hides the text. It is noted that this effect is mitigated in Firefox.

Overall, I think that the revised version improved and it will be a widely used resource for the community.

The paper is now suitable for publication, in my opinion.

Reviewer #3:

The reviewers' concerns and remarks have been addressed to my full satisfaction. The manuscript can be published as is.

26th Aug 2021

Manuscript number: MSB-2021-10387RR

Title: COVID-19 Disease Map, a computational knowledge repository of virus-host interaction mechanisms

Thank you again for sending us your revised manuscript. We are now satisfied with the modifications made and I am pleased to inform you that your paper has been accepted for publication.

*** PLEASE NOTE *** As part of the EMBO Publications transparent editorial process initiative (see our Editorial at <https://dx.doi.org/10.1038/msb.2010.72>), Molecular Systems Biology publishes online a Review Process File with each accepted manuscripts. This file will be published in conjunction with your paper and will include the anonymous referee reports, your point- by-point response and all pertinent correspondence relating to the manuscript. If you do NOT want this File to be published, please inform the editorial office at msb@embo.org within 14 days upon receipt of the present letter.

Should you be planning a Press Release on your article, please get in contact with msb@wiley.com as early as possible, in order to coordinate publication and release dates.

LICENSE AND PAYMENT:

All articles published in Molecular Systems Biology are fully open access: immediately and freely available to read, download and share.

Molecular Systems Biology charges an article processing charge (APC) to cover the publication costs. You, as the corresponding author for this manuscript, should have already received a quote with the article processing fee separately.

Please let us know in case this quote has not been received.

Once your article is at Wiley for editorial production you will receive an email from Wiley's Author Services system, which will ask you to log in and will present you with the publication license form for completion. Within the same system the publication fee can be paid by credit card, an invoice or pro forma can be requested.

Payment of the publication charge and the signed Open Access Agreement form must be received before the article can be published online.

Molecular Systems Biology articles are published under the Creative Commons licence CC BY, which facilitates the sharing of scientific information by reducing legal barriers, while mandating attribution of the source in accordance to standard scholarly practice.

Proofs will be forwarded to you within the next 2-3 weeks.

Thank you very much for submitting your work to Molecular Systems Biology.

Kind regards,
Jingyi

Jingyi Hou
Editor
Molecular Systems Biology

Corresponding Author Name: Marek Ostaszewski

Manuscript Number: MSB-2021-10387